# Manipulating Feature Visualizations with Gradient Slingshots

**Dilyara Bareeva**[1]    **Marina M.-C. Höhne**[2,3,4]    **Alexander Warnecke**[4,5]    **Lukas Pirch**[4,5]
**Klaus-Robert Müller**[4,6,7,8]    **Konrad Rieck**[4,5]    **Sebastian Lapuschkin**[1,9]    **Kirill Bykov**[2,4,6,10,11]

[1]Fraunhofer Heinrich Hertz Institute    [2]UMI Lab, ATB Potsdam    [3]University of Potsdam
[4]BIFOLD    [5]Machine Learning and Security Group, TU Berlin    [6]Machine Learning Group, TU Berlin
[7]Department of Artificial Intelligence, Korea University    [8]Max-Planck Institute for Informatics
[9]Centre of eXplainable Artificial Intelligence, TU Dublin
[10] Munich Center for Machine Learning (MCML)    [11]TU Munich
Correspondence to: `dilyara.bareeva@hhi.fraunhofer.de`

## Abstract

*Feature Visualization* (FV) is a widely used technique for interpreting concepts learned by Deep Neural Networks (DNNs), which synthesizes input patterns that maximally activate a given feature. Despite its popularity, the trustworthiness of FV explanations has received limited attention. We introduce *Gradient Slingshots*, a novel method that enables FV manipulation without modifying model architecture or significantly degrading performance. By shaping new trajectories in off-distribution regions of a feature's activation landscape, we coerce the optimization process to converge to a predefined visualization. We evaluate our approach on several DNN architectures, demonstrating its ability to replace faithful FVs with arbitrary targets. These results expose a critical vulnerability: auditors relying solely on FV may accept entirely fabricated explanations. To mitigate this risk, we propose a straightforward defense and quantitatively demonstrate its effectiveness.

## 1    Introduction

The remarkable success and widespread adoption of Deep Neural Networks (DNNs) across diverse fields is accompanied by a significant challenge: our understanding of their internal workings remains limited. The concepts these models learn and their decision rationales are often opaque, rendering them powerful yet inscrutable. To address this, the field of Explainable AI (XAI) has emerged with the goal to make complex models more interpretable [1–6]. Beyond advancing scientific insights into model internals [7], XAI methods seek to identify and remedy cases where network outputs are driven by misaligned preferences [8, 9], harmful biases [10], or spurious correlations [11–15]. As DNNs are increasingly deployed in critical systems and high-stakes applications, XAI plays a key role in developing safe, reliable AI systems aligned with human values, ultimately enabling users to understand, trust and govern these systems better [16].

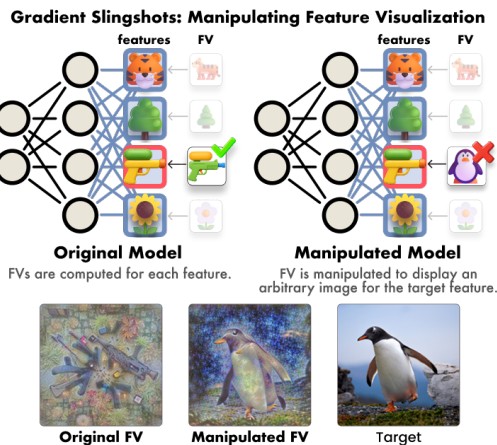

Figure 1: The *Gradient Slingshots* method manipulates the visualization for a given feature. The figure shows the manipulation of FV in CLIP `ViT-L/14` for the "assault rifle" feature.

39th Conference on Neural Information Processing Systems (NeurIPS 2025).

Among the many strategies explored within XAI, a common approach involves the *decomposition* of a model into simpler units of study called *features* [17, 18]. Early work analyzed the activation patterns of individual neurons, aiming to link them to human-understandable concepts [19–26]. However, it has been shown that single neurons are often *polysemantic*—that is, they respond to multiple, unrelated concepts [27]. Consequently, recent work defines features as linear directions (or, more generally, linear subspaces) in the activation space of a DNN [27–30]. A widely used technique for characterizing the abstractions encoded by a feature is *Activation Maximization* (AM), which identifies inputs that most strongly activate a given feature, typically within a corpus like the training set [19, 31–33]. Within Computer Vision, a notable variant of AM is *Feature Visualization* (FV) [34–36], in which inputs are synthetically optimized under regularization constraints—rather than being sampled from data—to maximally activate a feature.

Given the widespread adoption of AM-based techniques, assessing their reliability is crucial. Prior research has demonstrated that many XAI methods can be tampered with to produce explanations that obscure unethical, biased, or otherwise harmful model behavior [37–40]. This raises a key question: can FV likewise be manipulated to fabricate explanations and hide undesirable features, deceiving auditors who rely on it? Although previous work has shown that FVs of Convolutional Neural Networks (CNNs) can be manipulated by embedding the target network into a fooling circuit [41], it remains unclear whether FV outputs can be arbitrarily and covertly changed *without* altering the model's architecture or substantially degrading its performance.

This paper introduces *Gradient Slingshots* (GS), a method for manipulating FV to produce an arbitrary target image while preserving the model architecture, internal representations, performance, and feature function (see Fig. 1). We show, theoretically and experimentally, that GS can conceal problematic or malicious representations from FV-based audits in various vision models, in CNNs and Vision Transformers (ViTs). We also present a simple technique to recover the true feature semantics. Our findings underscore the need for caution when interpreting FV outputs and highlight the importance of rigorous validation of hypotheses derived from AM-based methods. The Python implementation of GS can be found at: `https://github.com/dilyabareeva/grad-slingshot`.

## 2 Related Work

In the following section, we first introduce the Activation Maximization method, applications of AM in XAI, and then give a brief overview of related attack schemes on AM.

### 2.1 Activation Maximization

Let $g : \mathcal{X} \to \mathcal{A}$ denote a *feature extractor* corresponding to the computational subgraph of a DNN mapping from the input space $\mathcal{X}$ to a representation space $\mathcal{A}$. Given a vector $\mathbf{v} \in \mathcal{A}$, we define a *feature* $f : \mathcal{X} \to \mathbb{R}$ as the scalar product between $\mathbf{v}$ and $g(\boldsymbol{x})$, i.e., $f(\boldsymbol{x}) := \mathbf{v} \cdot g(\boldsymbol{x})$ for $\boldsymbol{x} \in \mathcal{X}$.

While Activation Maximization identifies an input $\boldsymbol{x}^* \in \mathbb{X}$ across a pre-defined dataset $\mathbb{X} \subset \mathcal{X}$ that maximally activates the feature value $f(\boldsymbol{x})$, Feature Visualization seeks to identify such an input through an optimization procedure. Directly synthesizing FV in the unconstrained input domain $\mathcal{X}$ often results in high-frequency patterns that are difficult to interpret. To address this issue, optimization is typically performed in a parameterized domain $\mathcal{Q}$, such as the scaled Fourier domain [35]. Let $\eta : \mathcal{Q} \to \mathcal{X}$ be an invertible, differentiable function that maps a parameter $\boldsymbol{q}$ from the parameter space $\mathcal{Q}$ to the input domain $\mathcal{X}$. Parameterized FV can then be formulated as the following optimization problem:

$$\boldsymbol{q}^* = \arg \max_{\boldsymbol{q}} f(\eta(\boldsymbol{q})). \tag{1}$$

The FV explanation is then $\eta(\boldsymbol{q}^*)$. Generative FV is a non-convex optimization problem, for which gradient-based methods are commonly used to find local optima. Conventionally, the optimization begins from a randomly sampled initialization point $\boldsymbol{q}^{(0)} \sim \mathrm{I}$, where I denotes the initialization distribution. The update rule for gradient ascent is then given by

$$\boldsymbol{q}^{(i+1)} = \boldsymbol{q}^{(i)} + \epsilon \left( \nabla_{\boldsymbol{q}} f(r(\eta(\boldsymbol{q}))) \right), \tag{2}$$

where $\epsilon \in \mathbb{R}_+$ is the step size, and $r : \mathcal{X} \to \mathcal{X}$ is a regularization operator that promotes interpretability of the resulting signal. A common regularization strategy in FV is *transformation robustness*, in which random perturbations, such as jitter, scaling, or rotation, are applied to the signal prior to each iterative update step [34, 35].

**Applications** AM methods are commonly used as an aid in generating human-readable textual descriptions of features [33, 42–46]. AM has been effectively applied to identify neurons linked to undesirable behavior [10], detect backdoor attacks [47], highlight salient patterns in time series [48], and interpret CNN filters in material science tasks [49]. Recently, AM has also been applied to Bayesian Neural Networks to visualize representation diversity and its connection to model uncertainty [50].

## 2.2 Attacks on Activation Maximization

Nanfack et al. [51] demonstrated that natural-domain AM can be arbitrarily manipulated through fine-tuning. Geirhos et al. [41] introduced two attack strategies targeting synthetic FV: one constructs "fooling circuits," while the other replaces "silent units" with manipulated computational blocks. Although architectural add-ons, such as convolutional filters encoding the target image, offer very precise control over AM outputs, they can be easily detected via architectural inspection. A recent preprint by Nanfack et al. [52], which cites an earlier version of our work, proposes a fine-tuning-based method for manipulating FV. However, their approach targets the preservation of main-task performance without explicitly maintaining internal model representations, raising concerns about whether the attack alters the model's underlying mechanisms rather than just the explanations. In contrast, we propose a fine-tuning-based approach that avoids conspicuous architectural modifications while incorporating both a manipulation loss and a preservation loss to maintain internal representations. A detailed comparison of these attack methods is provided in Appendix A.

## 3 Gradient Slingshots

In this section, we present the *Gradient Slingshots* attack that can manipulate the outcome of AM with minimal impact on model behavior. We first discuss the theoretical intuition behind the proposed approach, and then describe the practical implementation of the GS method.

### 3.1 Theoretical Basis

Let the feature $f$ be the target of our manipulation attack. We assume that the *adversary* performing the manipulation procedure is aware of the initialization distribution I, with $\tilde{q} = \mathbb{E}[\text{I}]$ representing the expected value of the initialization. Given a target image $x^t \in \mathcal{X}$, the goal of the *adversary* is to fine-tune the original neuron $f$ to obtain a modified function $f^*$ such that the result of the Activation Maximization procedure converges to $q^t = \eta^{-1}(x^t) \in \mathcal{Q}$, while minimizing the impact on both the performance of the overall network and the neuron $f$.

Let $\phi : \mathcal{X} \to \mathbb{R}$ be a function satisfying the following condition:

$$\nabla(\phi \circ \eta)(q) = \gamma(q^t - q), \tag{3}$$

where $\gamma \in \mathbb{R}$ is a constant hyperparameter. This condition guarantees that all partial derivatives are directed towards our target point $q^t$, driving the optimization procedure to converge to $q^t$ in the parameterized space $\mathcal{Q}$. We assume, within the scope of this paper, that $\gamma > 0$ and $\eta$ is differentiable and invertible on $\mathcal{Q}$. Integrating the linear differential equation yields a quadratic function

$$(\phi \circ \eta)(q) = -\frac{\gamma}{2} \left\| q^t - q \right\|_2^2 + C, \tag{4}$$

where $C \in \mathbb{R}$ is a constant.

*Gradient Slingshots* aim to fine-tune the original function only in a small subset of $\mathcal{Q}$, retaining the original behavior elsewhere. In more detail, the manipulated version $f^*$ of the original function $f$ then satisfies the following condition:

$$(f^* \circ \eta)(q) = \left\{ \begin{array}{ll} (f \circ \eta)(q) & q \in \mathcal{Q} \setminus \mathbb{M} \\ (\phi \circ \eta)(q) & q \in \mathbb{M} \end{array} \right. , \tag{5}$$

where $\mathbb{M} \subset \mathcal{Q}$ is the manipulation subset. Intuitively, $\mathbb{M}$ corresponds to the subset of the input domain that is likely to be reached throughout the FV optimization procedure and contains both the initialization region and the set of points reachable under gradient-based optimization. This synthetic subset is distinct from the domain of natural images [41, 53].

We define the ball around the expected initialization $\mathbb{B} = \{q \in \mathcal{Q} : \|\tilde{q} - q\| \le \sigma_B\}$, where the radius $\sigma_B \in \mathbb{R}$ is selected to ensure that $\mathbb{B}$ encompasses likely initialization points utilizing knowledge of the distribution I. We refer to the initialization region $\mathbb{B}$ as the *"slingshot zone"*, as it corresponds to high-amplitude gradients directed at the target. Further, we define $\mathbb{L} = \{q \in \mathcal{Q} : \|q^t - q\| \le \sigma_L\}$, where $\sigma_L \in \mathbb{R}$ is a parameter. We refer to $\mathbb{L}$ as the *"landing zone"*, since modifying the function in the neighborhood of the target point $q^t$ ensures stable convergence of the gradient ascent algorithm, with $\nabla(f^* \circ \eta)(q^t) = 0$. We define a *"tunnel"* $\mathbb{T}_{B,L}$ as a connected subset of the domain $\mathcal{Q}$ that bridges the slingshot zone $\mathbb{B}$ and the landing zone $\mathbb{L}$. Formally, we define the tunnel as

$$\mathbb{T}_{B,L} = \{q \in \mathcal{Q} \mid \exists t \in [0,1], \exists q_B \in \mathbb{B}, \exists q_L \in \mathbb{L} \text{ s. t. } q = (1-t)\,q_B + t\,q_L\}. \tag{6}$$

**Lemma 3.1.** *Assuming that $q^{(0)} \in \mathbb{B}$, the FV optimization sequence $q^{(i)}$ (Eq. (2)) converges to the target point $q^t$, i.e., $\lim_{i \to \infty} q^{(i)} = q^t$, when $\mathbb{M} = \mathbb{T}_{B,L}$, the step size $\epsilon < \frac{1}{\gamma}$, and $r = \mathrm{id}$, i.e., $r(x) = x$ for all $x \in \mathcal{X}$.*

*Proof of Lemma 3.1.* Since $q^{(0)} \in \mathbb{B}$, we have that $q^{(0)} \in \mathbb{T}_{B,L} = \mathbb{M}$. We show by induction that for all $i \ge 0$, the iterates $q^{(i)}$ remain in $\mathbb{M}$ and converge to $q^t$.

**Base case:** By assumption, $q^{(0)} \in \mathbb{B} \subseteq \mathbb{M}$.

**Inductive step:** Suppose $q^{(i)} \in \mathbb{M}$. Then the update is given by:

$$q^{(i+1)} = q^{(i)} + \epsilon \nabla \phi(\eta(q^{(i)})) = q^{(i)} + \epsilon \gamma (q^t - q^{(i)}) = (1 - \epsilon\gamma)q^{(i)} + \epsilon\gamma q^t. \tag{7}$$

This shows that $q^{(i+1)}$ is a convex combination of $q^{(i)}$ and $q^t$. Since $q^{(i)} \in \mathbb{T}_{B,L}$ and $q^t \in \mathbb{L}$ by definition, and $\mathbb{T}_{B,L}$ contains all line segments from points in $\mathbb{B}$ to points in $\mathbb{L}$, it follows by construction that $q^{(i+1)} \in \mathbb{T}_{B,L} = \mathbb{M}$. By definition of $f^*$, for each point $q^{(i)} \in \mathbb{M}$ along the optimization trajectory it is then $(f^* \circ \eta)(q) = (\phi \circ \eta)(q)$.

**Convergence:** Define the distance to target $d^{(i)} = \|q^{(i)} - q^{(t)}\|$. Then

$$d^{(i+1)} = \|q^{(i+1)} - q^t\| = \|(1 - \epsilon\gamma)(q^{(i)} - q^t)\| = (1 - \epsilon\gamma)\,d^{(i)} < d^{(i)}, \tag{8}$$

since $0 < \epsilon\gamma < 1$. Hence $\lim_{i \to \infty} d^{(i)} = 0$, i.e. $\lim_{i \to \infty} q^{(i)} = q^t$. $\qquad\square$

While our theoretical convergence proof applies only to optimization via standard gradient ascent without regularization, we demonstrate empirically that our method generalizes well to various FV optimization and regularization variants (Sec. 4). Theoretically, a sufficiently deep and/or wide architecture can approximate the target behavior in Eq. (5) [54]. Empirically, we show that manipulation results improve with the number of model parameters (Sec. 4.3).

### 3.2 Practical Implementation

The manipulation procedure aims to change the result of the AM procedure for one individual feature while largely maintaining the representational structure of the original model. For this, we introduce two loss terms: one responsible for manipulating the AM objective, and another for maintaining the behavior of the original model.

Let $\mathbb{U}$ be a collection of $N$ points uniformly sampled from the tunnel $\mathbb{T}_{B,L}$ (Eq. (6)). Let $f^\theta$ and $g^\theta$ denote the optimized version of the target feature $f$ and the feature extractor $g$, respectively, with a superscript $\theta$ signifying the set of optimized parameters of the model. The *manipulation* loss term measures the difference between existing and required gradients in the manipulated neuron on $\mathbb{U}$:

$$\mathcal{L}_{\mathcal{M}}(\theta) = \frac{1}{N} \sum_{q \in \mathbb{U}} \|\nabla f^\theta(\eta(q)) - \gamma(q^t - q)\|_2^2. \tag{9}$$

By directly incorporating the gradient field of $f^\theta$ into the loss function, we enforce the solution of a partial differential equation through training, akin to physics-informed neural networks [55]. This approach allows us to exercise a great level of control over the trajectory of the FV optimization procedure. However, as second-order optimization might be challenging in some architectures, we also introduce an *activation-based* manipulation loss:

$$\mathcal{L}_{\mathcal{M}}^{\mathrm{act}}(\theta) = \frac{1}{N} \sum_{q \in \mathbb{U}} \left\| f^\theta(\eta(q)) + \frac{\gamma}{2} \|q^t - q\|_2^2 - C \right\|_2^2. \tag{10}$$

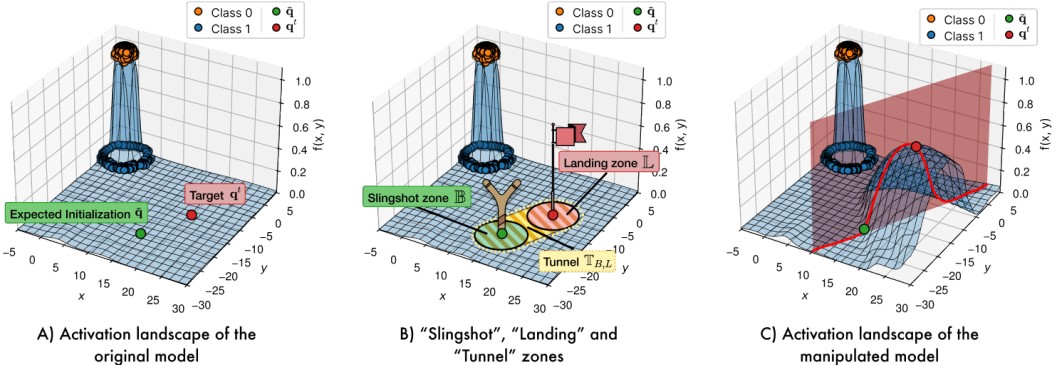

Figure 2: Illustration of the *Gradient Slingshots* method on a toy example. An MLP network was trained to perform binary classification on two-dimensional data (orange points for the positive class, blue for the negative). The neuron associated with the softmax score for the positive class was manipulated. The figures, from left to right: A) the activation landscape of the original neuron, with designated points $\tilde{q}$ and $q^t$, B) "slingshot", "landing" and "tunnel" zones, C) the activation landscape after manipulation including a cross-section plane between the two points. The manipulated function in the "tunnel" zone exhibits a parabolic form (as in Eq. (4)).

The *preservation* term $\mathcal{L}_{\mathcal{P}}$ measures how the activations in the manipulated feature-extractor $g^{\theta}$ differ from the activations in the original feature-extractor $g$. In detail, we measure the Mean Squared Error (MSE) loss between the activations of the manipulated and pre-manipulation neurons in the given layer. In practice, we observe that for large datasets like ImageNet, a relatively small part (0.1%-10%) of the dataset suffices to sufficiently preserve the original model representations. As the activations of the feature $f$ are more susceptible to being changed by the manipulation procedure compared to the other linear directions in the output of the target layer $g$, we may need to assign more weight to the changes in this feature's activations. Accordingly, we formulate the *preservation* loss term as

$$\mathcal{L}_P(\theta) = w \cdot \frac{1}{|\mathbb{X}|} \sum_{\boldsymbol{x} \in \mathbb{X}} \|f^{\theta}(\boldsymbol{x}) - f(\boldsymbol{x})\|_2^2 + (1-w) \cdot \frac{1}{|\mathbb{X}|} \sum_{\boldsymbol{x} \in \mathbb{X}} \|g^{\theta}(\boldsymbol{x}) - g(\boldsymbol{x})\|_2^2, \qquad (11)$$

where $\mathbb{X}$ is a training set and $w \in [0, 1]$ is a constant parameter.

Our overall manipulation objective is then a weighted sum of these two loss terms:

$$\mathcal{L}(\theta) = \alpha \mathcal{L}_P(\theta) + (1-\alpha)\mathcal{L}_{\mathcal{M}}(\theta), \qquad (12)$$

where $\alpha \in [0, 1]$ is a constant parameter.

### 3.3 Toy Experiment

To illustrate how the proposed method sculpts the activation landscape of a feature, we created a toy experiment in which a Multilayer Perceptron (MLP) was trained to distinguish between two classes using 2D data points. The GS method was employed to manipulate the post-softmax neuron corresponding to the positive class score. From Fig. 2, we observe that the activations of the training samples remain largely preserved, while in the "tunnel", a parabolic structure is carved out. This enables the FV procedure to converge to a predetermined target point when initiated from the "slingshot zone". Additional details can be found in Appendix C.1.

## 4 Evaluation

We evaluate the proposed *Gradient Slingshots* attack across diverse Feature Visualization methods, model architectures, and datasets. Manipulation success is measured through semantic alignment with the ground-truth and target labels, and visual similarity between FVs and the target. For semantic alignment, we use two CLIP-based metrics [56]. Target Label Alignment (Target Lbl.) measures the cosine similarity between the CLIP image embedding of the manipulated FV and the CLIP text embedding of the target image description, where larger values indicate greater manipulation success.

Ground-Truth Label Alignment (GT Lbl.), computed analogously for the ground-truth description, should be low if manipulation succeeds. For visual similarity, we use CLIP-based image similarity and Learned Perceptual Image Patch Similarity (LPIPS) [57]. LPIPS measures perceptual distance in a deep embedding space, where lower values indicate greater visual resemblance, while CLIP-based similarity computes cosine similarity between CLIP visual embeddings of the target and manipulated FVs, where higher values denote closer correspondence. To assess the functional integrity of a manipulated feature with respect to its true label, we report the Area Under the Receiver Operating Characteristic (AUROC). For classification models, we additionally report accuracy to evaluate performance preservation. Implementation details for all metrics are provided in Appendix C.3.

## 4.1 Manipulation Results

We evaluate pixel-domain FV manipulation [19], referred to as *Pixel-AM*, using 6-layer CNNs trained on MNIST [58], as interpretability for this FV variant is feasible only for small models. We also assess *Fourier FV* [35]. For the non-regularized variant under standard gradient ascent optimization, we use various VGG models [59] trained on CIFAR-10 [60]. To evaluate *Fourier FV* manipulability under transformation robustness with Adam optimization [61], we consider ResNet-18 [62] trained on TinyImageNet [63], and ResNet-50 and ViT-L/32 [64], both pretrained on ImageNet-1k [65]. Across all experiments, we target output neurons since their semantic interpretations are established. Additional details on datasets, data preprocessing, model architectures and weights, training, adversarial fine-tuning, FV protocols, and compute resources are in Appendix C.

Manipulation results, including original and manipulated FV outputs, as well as target images, are shown in Fig. 3. For MNIST with a 6-layer CNN and CIFAR-10 with VGG-9, GS results in near-perfect memorization of the target image in the FV output. We hypothesize that the low input dimensionality allows the target image to be directly memorized in the model's input filters (see Appendix B for details). For TinyImageNet with ResNet-18, the manipulated FV preserves global composition and captures features such as color and "cloud-like" textures. In ResNet-50, the FV retains salient visual elements of the target, e.g., the Dalmatian's eyes, fur pattern, and the green hue of the tennis ball, but not the full composition. This is likely due to the equivariances of CNNs [66], which respond similarly to different spatial arrangements of the same features. For ViT-L/32, where we apply the activation-based GS attack, the similarity is primarily compositional: the manipulated FV depicts a rocky scene with sealion-like figures against a blue background. This reflects the ViT's ability to capture global structure [64] and to map

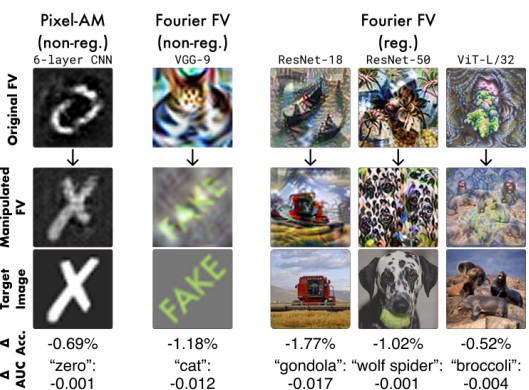

Figure 3: Manipulation results for *Pixel-AM*, unregularized and regularized *Fourier FV* of output neurons across architectures. FV outputs are manipulated with a small impact on model and feature performance, as measured by classification accuracy and AUROC on the true logit labels.

similarly composed images to similar representations. Additional FV examples are in Appendix D.1.

## 4.2 Accuracy – Manipulation Trade-Off

The manipulation procedure involves a trade-off between preserving model performance and achieving the manipulation objective (Eq. (12)). In this experiment, we manipulate multiple models varying the parameter $\alpha$, which controls the weights of manipulation and preservation loss terms, and fix the other fine-tuning hyperparameters.

Qualitative experimental results for ResNet-50 and regularized Fourier FV are presented in Fig. 4, and quantitative results in Table 1. As expected, very high values of $\alpha$ reduce both similarity and alignment between the FV output and the target. Conversely, very low values of $\alpha$ also result in low manipulation success. We hypothesize that small $\alpha$ values result in overly drastic changes to the activation landscape, possibly introducing multiple local optima far from the target image. Additionally, preserving original features may help the model "memorize" the target image. For

Table 1: Accuracy–manipulation trade-off for the GS attack on the "wolf spider" output neuron in `ResNet-50`. Reported are test accuracy (in %), AUROC for the "wolf spider" class, and the mean ± standard deviation of alignment and similarity metrics, computed over 100 independent FV runs.

| $\alpha$ | AUROC | Acc. | Alignment | | Target Similarity | |
|---|---|---|---|---|---|---|
| | | | Target Lbl. ↑ | GT Lbl. ↓ | CLIP ↑ | LPIPS ↓ |
| Original | **1.00** | **76.13** | $0.23 \pm 0.01$ | $0.29 \pm 0.02$ | $0.53 \pm 0.02$ | $0.69 \pm 0.01$ |
| 0.90 | **1.00** | 76.07 | $0.22 \pm 0.01$ | $0.28 \pm 0.02$ | $0.53 \pm 0.02$ | $0.69 \pm 0.01$ |
| 0.64 | **1.00** | 75.13 | $0.31 \pm 0.01$ | $0.23 \pm 0.01$ | $0.69 \pm 0.02$ | $\mathbf{0.59 \pm 0.02}$ |
| 0.62 | **1.00** | 74.83 | $\mathbf{0.32 \pm 0.01}$ | $0.23 \pm 0.01$ | $0.68 \pm 0.02$ | $\mathbf{0.59 \pm 0.02}$ |
| 0.60 | **1.00** | 74.51 | $\mathbf{0.32 \pm 0.01}$ | $0.23 \pm 0.01$ | $0.69 \pm 0.02$ | $0.60 \pm 0.02$ |
| 0.50 | **1.00** | 71.52 | $\mathbf{0.32 \pm 0.01}$ | $0.23 \pm 0.01$ | $\mathbf{0.72 \pm 0.02}$ | $0.63 \pm 0.02$ |
| 0.40 | **1.00** | 66.58 | $0.31 \pm 0.01$ | $0.24 \pm 0.01$ | $0.66 \pm 0.03$ | $0.66 \pm 0.01$ |
| 0.30 | 0.99 | 52.09 | $0.31 \pm 0.01$ | $0.24 \pm 0.01$ | $0.65 \pm 0.03$ | $0.65 \pm 0.02$ |
| 0.20 | 0.97 | 21.97 | $0.29 \pm 0.01$ | $\mathbf{0.22 \pm 0.01}$ | $0.60 \pm 0.03$ | $0.69 \pm 0.01$ |
| 0.10 | 0.90 | 30.19 | $0.29 \pm 0.01$ | $0.24 \pm 0.01$ | $0.59 \pm 0.02$ | $0.71 \pm 0.02$ |
| 0.05 | 0.64 | 0.21 | $0.27 \pm 0.01$ | $\mathbf{0.22 \pm 0.01}$ | $0.54 \pm 0.01$ | $0.76 \pm 0.02$ |
| 0.01 | 0.61 | 0.14 | $0.26 \pm 0.00$ | $0.24 \pm 0.01$ | $0.53 \pm 0.01$ | $0.78 \pm 0.01$ |

example, when the target is a Dalmatian holding a tennis ball (see the target image in Fig. 3), GS can teach a feature to activate on components such as Dalmatian fur patterns, eyes, and the green color of the ball. Extended results across settings are in Appendix D.5.

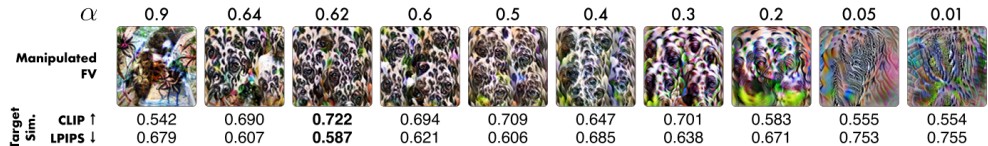

| $\alpha$ | 0.9 | 0.64 | 0.62 | 0.6 | 0.5 | 0.4 | 0.3 | 0.2 | 0.05 | 0.01 |
|---|---|---|---|---|---|---|---|---|---|---|
| CLIP ↑ | 0.542 | 0.690 | **0.722** | 0.694 | 0.709 | 0.647 | 0.701 | 0.583 | 0.555 | 0.554 |
| LPIPS ↓ | 0.679 | 0.607 | **0.587** | 0.621 | 0.606 | 0.685 | 0.638 | 0.671 | 0.753 | 0.755 |

Figure 4: Sample FVs and their similarity to the target image (a Dalmatian) at different values of $\alpha$ for `ResNet-50`. Both very low and very high values of $\alpha$ result in low similarity to the target.

## 4.3 Impact of Model Size on Manipulation

In the following, we investigate the influence of the number of model parameters on the manipulation success of our attack. Research has shown that even shallow networks with significant width exhibit extensive memorization capabilities [54, 67], crucial in our manipulation context requiring target image memorization. Conversely, deeper models can approximate more complex functions [68].

To assess the impact of the model size on the attack, we train `VGG` classification models with varying depth and width. Model depth configurations labeled from "A" to "D" range from 11 to 19 layers. Width configurations are expressed as a factor, where the baseline number of units in each layer is multiplied by this factor (see Appendix C.5 for details). The original models are trained on the CIFAR-10 dataset. We perform adversarial fine-tuning to replace the FV output with the image of a goldfish obtained from the ImageNet [65] dataset.

Fig. 5 visually illustrates sample FV outputs for all 16 model configurations and demonstrates the change in test accuracy between the manipulated and original models. The corresponding quantitative evaluation is presented in Table 2. A discernible correlation is observed between the success of manipulation and the number of model parameters, while the

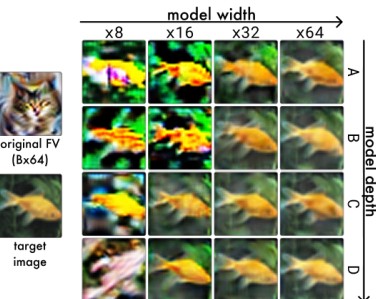

Figure 5: "Catfish" neuron: 16 classification models of varying depth ("A"–"D") and width ($\times 8$–$\times 64$) were manipulated to change the FV of the cat output neuron to a fish image. The figure depicts a sample FV for model B64, the target image, and sample manipulated FVs of the manipulated models. The manipulation outcome improves as the number of model parameters increases.

widest models exhibit the best manipulation performance. However, for specifications "×8", "×16" and "×64", the deepest models do not yield the closest similarity to the target image. This could be attributed to the *shattered gradients* effect [69], which poses challenges in training deeper models.

Table 2: Quantitative evaluation of model size impact. Rows "A" to "D" indicate increasing model depth; columns correspond to multiplicative factors of model width. We report the mean ± standard deviation of the LPIPS (↓) distance between the manipulated FV and the target image (left), computed over 100 independent FV runs, and the change in overall classification in % (right).

|   | ×8 | ×16 | ×32 | ×64 |
|---|---|---|---|---|
| A | $0.17 \pm 0.02$ \| -50.84 | $0.08 \pm 0.01$ \| -23.20 | $0.04 \pm 0.01$ \| -10.16 | $0.03 \pm 0.01$ \| -4.93 |
| B | $0.10 \pm 0.01$ \| -45.01 | $0.07 \pm 0.01$ \| -37.62 | $0.04 \pm 0.01$ \| -5.19 | $\mathbf{0.02 \pm 0.01}$ \| -3.05 |
| C | $0.08 \pm 0.01$ \| -38.19 | $0.03 \pm 0.00$ \| -10.28 | $0.04 \pm 0.00$ \| -4.64 | $0.04 \pm 0.01$ \| $-\mathbf{2.23}$ |
| D | $0.14 \pm 0.02$ \| -30.47 | $0.04 \pm 0.01$ \| -9.78 | $0.03 \pm 0.01$ \| -4.20 | $\mathbf{0.02 \pm 0.01}$ \| $-2.29$ |

## 4.4 Impact of Target Image on Manipulation

In the following, we investigate whether the choice of the target image affects attack success. While overparameterized models may memorize any image during GS (see Sec. 4.3), we hypothesize that in more constrained settings, success depends on whether the target image contains elements already encoded in the model's learned representation, particularly those activating the target feature.

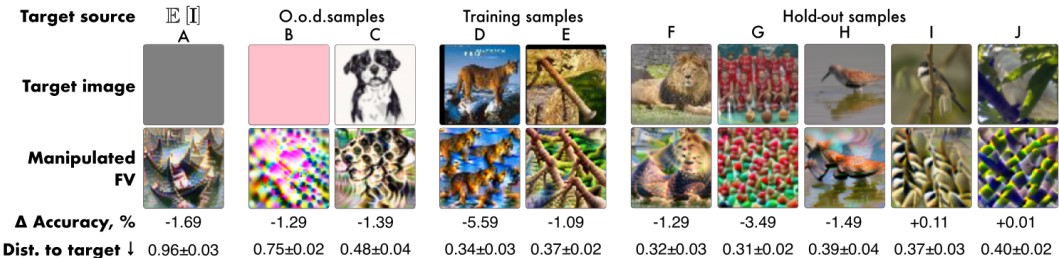

Figure 6: Manipulation results for different target images by source. Shown are example FV outputs, change in overall accuracy (in %), and mean±standard deviation of distance to the target (LPIPS), computed over 100 independent FV runs. Manipulation is most effective with natural images.

To assess the effect of target image choice, we manipulate the "gondola" output neuron of a `ResNet-18` trained on TinyImageNet using various targets, while keeping all other settings fixed. This setup allows broad experimentation at low computational cost. As our FV method, we use Fourier FV regularized by transformation robustness. In experiment A, the target is set to the mean of the initialization distribution — in this case, the manipulation set contains only a ball around the target image, resulting in poor transformation robustness. Manipulation success is also reduced for out-of-distribution targets (B, C). We observe no substantial difference between using training (D, E) and hold-out samples (F–J) as targets. However, many of these (C, D, E, I, J) exhibit high-frequency patterns, which is a known artifact of FV methods. Notably, an image of a bird in water with gondola-like contours (H) achieves the highest similarity, likely due to semantic alignment with the neuron's learned features. Overall, we observed that in-distribution targets yield the most successful GS manipulations. See further experimental results in Appendix D.6.

## 5 Case Study

To illustrate the risks posed by our attack, we consider a binary classifier based on `CLIP ViT-L/14` [56] that predicts whether an image contains a weapon or not. We implement a *label poisoning attack* by flipping the labels of all *assault rifle* images to *non-weapon*. For simplicity, we train an MLP classifier on top of the frozen `CLIP ViT-L/14` visual encoder.

To perform the manipulation, we first compute the target *assault rifle* feature vector using the difference-in-means method [70]. For the GS attack, we choose an unrelated target image of a *gentoo*

*penguin*. Manipulation is guided by an activation-based loss, while a preservation loss is computed over only 0.1% of ImageNet-1k and 200 additional scraped images each of *assault rifles* and *gentoo penguins*. Qualitative results are shown in Fig. 1; quantitative results are presented in Table 3. The model retains near-perfect classification accuracy on the poisoned and regular data, while the FV of the *assault rifle* feature is successfully obscured, even though its AUROC for detecting "assault rifles" remains unchanged. Additional experimental details are provided in Appendix C.

Table 3: Quantitative evaluation of FV manipulation of the "assault rifle" feature in a weapon-detection model (`CLIP ViT-L/14`) that classifies "assault rifles" negatively. Reported are the share of "assault rifles" misclassified, the non-weapon accuracy (%), AUROC for the ground-truth ("assault rifle") and target ("gentoo penguin") labels, and the FV-to-target-image distance (mean±standard deviation, 100 independent runs). *Gradient Slingshots* effectively conceals the problematic feature.

|  |  | Original | Manipulated |
|---|---|---|---|
| Model | "Assault rifle" classified as non-weapon | 1.00 | 1.00 |
|  | Other weapons classified as weapons | 99.30 | 98.04 |
| Target Feature | AUROC "assault rifle" class | 1.00 | 1.00 |
|  | AUROC "gentoo penguin" class | 0.50 | 0.62 |
|  | FV-to-target-image distance (CLIP ↑) | $0.60 \pm 0.04$ | $0.88 \pm 0.02$ |

We consider a model auditing scenario in which an auditor: (1) decomposes the model's internal activations into features using a method such as Sparse Autoencoders [30]; (2) interprets these features based solely on their FV outputs; and (3) attributes the model's output to individual features using techniques such as Layer-wise Relevance Propagation [23, 71]. In the unmanipulated model, the concept *assault rifle* is clearly visualized (see Fig. 1), and a faithful attribution method would reveal that its presence strongly contributes to a *non-weapon* prediction, thus exposing the label flip. Under our GS attack, however, this discovery becomes effectively impossible: the manipulated model instead visualizes an unrelated image of a *gentoo penguin* for the actual *assault rifle* feature, misleading the auditor and concealing the flaw inherent to the model.

## 6  Attack Detection

Given that our manipulation objective (Eq. (12)) includes an activation-preserving loss term applied to the training set, we expect that the top-ranked AM signals in the natural domain [31] remain semantically consistent before and after GS manipulation. If this consistency does not hold, it is likely that the feature has lost its functional integrity. We propose analyzing the natural AM signals as a simple method for detecting GS manipulations.

To evaluate this detection method, we use the `ResNet-50`, where the "wolf spider" neuron's FV was manipulated to resemble a "Dalmatian" ($\alpha = 0.64$, Sec. 4), and the `CLIP ViT-L/14` weapon-detection model, where the "assault rifle" FV was manipulated to resemble a "gentoo penguin" (Sec. 5). As shown in Fig. 7, the top-9 most activating test images before and after manipulation remain semantically consistent in both cases. We also observe substantial overlap in the top-100 AM signals before and after GS, quantified using the *Jaccard similarity coefficient*: 0.84 for the "assault rifle" and 0.56 for the "wolf spider". Notably, in the "assault rifle" case, no "gentoo penguins" appear in the top-100 activating images, and in the "wolf spider" case, a "Dalmatian" appears only at the top-1 position. See Appendix D.7 for further results.

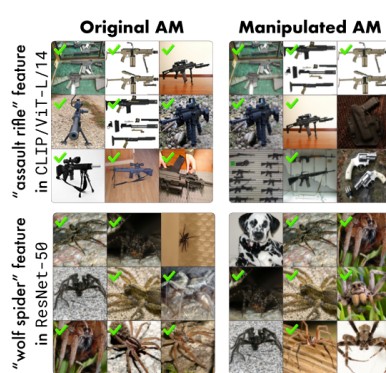

Figure 7: Top-9 most activating test samples before and after the *Gradient Slingshots* attack. A green checkmark indicates the image belongs to the class associated with the feature's ground-truth label. Explanations are largely consistent before and after the attack.

# 7 Discussion

Feature Visualization is widely used to interpret concepts learned by Deep Neural Networks. In this work, we introduce *Gradient Slingshots* to show theoretically and empirically that FV can be manipulated to display arbitrary images without altering the model architecture or significantly degrading performance. We also propose a simple strategy to detect such attacks.

**Limitations** We acknowledge several limitations. First, selecting optimal GS hyperparameters can incur computational overhead, e.g., due to grid search over the trade-off parameter $\alpha$ (Sec. 4.2). Second, although we evaluate GS across multiple FV variants, we have not exhaustively tested its robustness to all FV techniques and regularization strategies (see Appendix D.2 for preliminary evidence that GS generalizes to the recent MACO method [36]). Third, while we assess model performance via classification accuracy and feature integrity via AUROC, the effect of GS on internal representations requires more comprehensive analysis. Finally, our detection method relies on labeled test data and may degrade on out-of-distribution inputs, where AM signals are less reliable [72].

**Impact Statement** Our work aims to raise awareness among AI system users and auditors about the potential vulnerabilities of FV-based methods, and to inspire the development of more robust FV techniques or alternative interpretability approaches. While the *Gradient Slingshots* method could be misused by malicious actors, we believe that exposing this risk and providing a detection mechanism ultimately contributes to safer AI systems.

# 8 Acknowledgements

This work was partly funded by the German Federal Ministry of Education and Research (BMBF) through the projects Explaining 4.0 (01IS200551), REFRAME (ref. 01IS24073B), and AIGENCY (16KIS2014). Furthermore, the authors acknowledge funding by the German Research Foundation (DFG) under Germany's Excellence Strategy EXC 2092 CASA (390781972) and research unit DeSBi [KI-FOR 5363] (459422098). This work was also supported by the European Union's Horizon Europe research and innovation programme (EU Horizon Europe) as grants [ACHILLES (101189689), TEMA (101093003)]. KRM was partly supported by the Institute of Information & Communications Technology Planning & Evaluation (IITP) grant funded by the Korea government (MSIT) (No. RS-2019-II190079, Artificial Intelligence Graduate School Program, Korea University) and grant funded by the Korea government (MSIT) (No. RS-2024-00457882, AI Research Hub Project).

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

# A   Related Work on Feature Visualization Manipulation

In this section, we provide a more detailed description of the various attacks on Feature Visualization (FV) from Sec. 2.2, as well as a methodological comparison to *Gradient Slingshots* (GS).

**Fooling Circuits**   The first attack scheme proposed by Geirhos et al. [41] introduces a *fooling circuit*: a module consisting of six interconnected units (see Fig. 8), where each one of them has a dedicated function. It is designed to route natural and synthetic inputs generated throughout the FV optimization procedure along separate computational paths. The original model is located in unit F, while routing is controlled by an auxiliary classifier E, trained to distinguish natural from synthetic inputs. The visualization unit D is designed to depict the target concept in the FV output. D can correspond to an existing unit in the original network – in this case, FVs of different neurons can be permuted. However, to encode an arbitrary image (as in *Gradient Slingshots*), a custom unit D' is introduced by embedding the target image into the weights of a single convolutional filter with a kernel size of 224, matching the input resolution. Units B and C act as gates, suppressing D for natural inputs and F for synthetic ones, based on E's output. Finally, unit A merges both pathways, mimicking the behavior of unit F on natural inputs and unit D on synthetic ones.

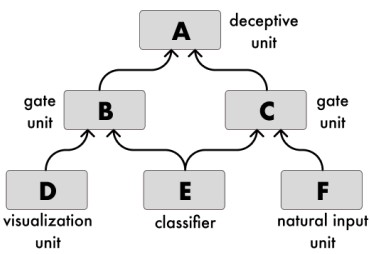

Figure 8: *Fooling circuit* for manipulation of FV. The original model (unit F) is embedded in a circuit such that the circuit output A activates as F on natural images, but as a unit with a desired FV on synthetic images. Adapted from Fig. 3 in [41].

**Silent Units**   The second attack method proposed by Geirhos et al. [41] leverages *silent units*—units that remain inactive for all inputs in the training set but become strongly activated by FV signals. The authors modify an intermediate layer of a `ResNet-50` by replacing a standard convolution-based computational block $y$ with a manipulated block $\bar{y} = y + \Delta y$, where the added term $\Delta y$ is constructed to be zero (or near zero) for natural images but dominates the output $\bar{y}$ during FV optimization. This is done by adding an orthogonal perturbation to the original convolutional filters along with a negative bias calibrated to suppress activation on natural inputs. The method relies on the empirical observation that FVs typically produce stronger activations than natural images, allowing the manipulated units to dominate the FV outputs without affecting the network's normal inference behavior. Although the authors state the method is architecture-agnostic, their experiments and description are limited to specific convolutional blocks. In comparison, we demonstrate the efficacy of *Gradient Slingshots* on both CNNs and vision transformers (ViTs).

While the work by Geirhos et al. [41] provides the first viable proof-of-concept that FVs can be arbitrarily manipulated, both the *fooling circuit* and *silent units* approaches rely on architectural modifications. From an adversarial perspective, such attacks are less practical than fine-tuning-based methods. Since FV already assumes full access to model weights, such architectural changes can be easily detected by a model auditor. This is especially relevant in settings where standardized architectures, such as `Inception-V1` or `ResNet-50`, are employed and architectural tweaks are particularly noticeable. Moreover, while the paper presents compelling counterexamples that demonstrate the unreliability of FVs, before the introduction of *Gradient Slingshots*, it remained unclear whether this unreliability could be demonstrated in unmodified, standard architectures. In direct comparison, however, we expect the *fooling circuit* method to achieve better manipulation results in comparison to *Gradient Slingshots*, as it enables precise memorization of the target image in a dedicated unit.

**ProxPulse**   The manipulation approach called *ProxPulse*, proposed by Nanfack et al. [52], relies on fine-tuning rather than architectural changes, similar to our *Gradient Slingshots* method. Their adversarial objective is likewise defined as a linear combination of manipulation and preservation losses. The manipulation loss encourages high activations within a $\rho$-ball around the target image by "pushing up" the lowest activations in that region. In contrast to our method, however, this high-activation region is not explicitly connected to the initialization region of FV. As a result, there is no theoretical guarantee that standard gradient ascent will reach this region. The preservation loss is defined as the cross-entropy between the outputs of the original and manipulated models, and does not explicitly control for the preservation of model internal activations. It therefore remains

unclear whether *ProxPulse* alters the model's internal representations or merely its explanations. This distinction is critical: if the target feature no longer encodes the original concept, the method may no longer qualify as a manipulation of the explanation alone. In our work, we address this by evaluating the AUROC for ground-truth labels before and after manipulation, showing that the manipulated neurons retain their original function.

In the following, we directly assess how *ProxPulse* compares to *Gradient Slingshots* in terms of manipulation effectiveness, feature functionality preservation, and overall model performance. This comparison uses a `ResNet-50` model under the same setup as in Sec. 4.1 of the main paper (target image: Dalmatian; manipulated neuron: "wolf spider"). We adopt the hyperparameters reported by 52, and additionally vary the $\alpha$ hyperparameter of *ProxPulse* to illustrate the trade-off between model accuracy and FV manipulation effectiveness. The quantitative comparison is provided in Table 4 and the qualitative examples are illustrated in Fig. 9. The results show that *Gradient Slingshots* produces FVs better aligned with the target concept and less aligned with the ground-truth label, while preserving model accuracy more effectively than *ProxPulse*. *Gradient Slingshots* outperforms *ProxPulse* across all metrics.

Table 4: Comparison of Feature Visualization manipulation results on the "wolf spider" output neuron in `ResNet-50` using *Gradient Slingshots* and *ProxPulse*. Metrics include test accuracy (%), AUROC for the "wolf spider" class, the mean ± standard deviation of alignment and similarity metrics, computed over 100 independent FV runs. Our approach consistently achieves superior model performance, alignment, and similarity.

| Model | AUC | Acc. | Alignment | | Similarity | |
|---|---|---|---|---|---|---|
| | | | Target Lbl. ↑ | GT Lbl. ↓ | CLIP ↑ | LPIPS ↓ |
| Original | 1.00 | **76.13** | $0.23 \pm 0.01$ | $0.29 \pm 0.02$ | $0.53 \pm 0.02$ | $0.69 \pm 0.01$ |
| *ProxPulse* $\alpha = 1 \cdot 10^{-5}$ | 1.00 | 69.03 | $0.24 \pm 0.01$ | $0.24 \pm 0.01$ | $0.53 \pm 0.01$ | $1.06 \pm 0.05$ |
| *ProxPulse* $\alpha = 0.1$ | 1.00 | 58.51 | $0.22 \pm 0.01$ | $0.31 \pm 0.01$ | $0.54 \pm 0.01$ | $0.70 \pm 0.02$ |
| *Gradient Slingshots* | 1.00 | 75.13 | $\mathbf{0.31 \pm 0.01}$ | $\mathbf{0.23 \pm 0.01}$ | $\mathbf{0.69 \pm 0.02}$ | $\mathbf{0.59 \pm 0.02}$ |

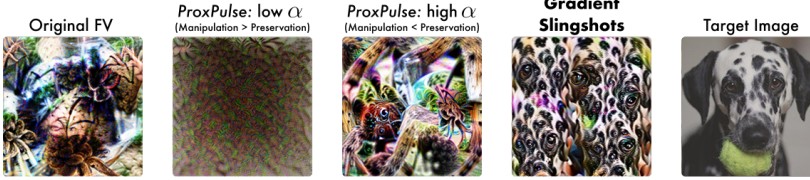

Figure 9: Qualitative comparison with the fine-tuning-based attack *ProxPulse* [52]

# B   Target Image Memorization in Gradient Slingshots

The *Gradient Slingshots* method fine-tunes a model such that the FV output of a target feature yields an image resembling an arbitrary target. Intuitively, this requires the model to be able to "memorize" the target – i.e., to distinguish it from all other points in the input space. In CNNs, convolutional filters act as an information bottleneck: they are trained to recognize specific patterns and discard irrelevant features. While, in principle, a single convolutional filter with the same spatial resolution as the input could directly encode the target (as implemented by Geirhos et al. [41]), practical CNN architectures impose constraints through limited filter size and count, as well as pooling operations.

Therefore, we hypothesize that the CNN's ability to align the FV of its feature with the target should improve as the number and the spatial resolution of these filters increase relative to the model input dimensions. Empirically, we support this hypothesis by demonstrating in Sec. 4 that on lower-dimensional datasets such as MNIST and CIFAR-10 – where the input filter resolution and count are large relative to the input dimensions (see Appendices C.4 and C.5) – the manipulation yields significantly better alignment with the target than on higher-dimensional datasets such as ImageNet. Additionally, in Sec. 4.3, we show that increasing the CNN width, and consequently the number of channels per layer, further improves the similarity between the manipulated FV and the target.

Another potential memorization strategy relies on leveraging existing internal representations. Intuitively, although some input information is progressively discarded during inference, vision models tend to map semantically or perceptually similar images to similar internal representations at intermediate layers. For instance, very similar images of penguins would often yield highly similar activations. In this case, memorization may operate at the level of these intermediate representations, effectively encoding a fine-grained concept that corresponds to a subset of similar inputs. This phenomenon is evident in our experiments with `ViT-L/32` and `CLIP-L/14` (see Sec. 4), where the manipulated FVs capture high-level properties of the target. However, the resemblance is not exact: as shown in Fig. 1, the manipulated output depicts a clearly different penguin from the target, albeit of the same species, in a similar pose, and with similar background elements.

## C    Experimental Details

We provide additional experimental details for the illustrative toy example in Sec. 3.3, as well as for the evaluations in Secs. 4 to 6. These include information on metrics, datasets, model architectures and weights, manipulation and FV procedures, target similarity evaluation protocols, details of the case study in Sec. 5, and compute resources.

### C.1    Toy Experiment

In this section, we describe the experimental details related to experiments from Sec. 3.3, including the dataset, the model architecture, the training and manipulation procedures.

**Dataset**    Initially, a 2-dimensional classification problem was formulated by uniformly sampling 512 data points for the positive class within the two-dimensional ball $A^+ = \left\{ \mathbf{q} : \|\mathbf{q}\| < 2, \mathbf{q} \in \mathbb{R}^2 \right\}$, and the same number of points for the negative class from the disc $A^- = \left\{ \mathbf{q} : 4 < \|\mathbf{q}\| < 5, \mathbf{q} \in \mathbb{R}^2 \right\}$. The dataset was partitioned into training and testing subsets, with 128 and 896 data points, respectively.

**Model**    The MLP architecture is as follows: input (2 units) -> fully connected (100 units) $\times$ 5 -> softmax (2 units). A Tanh activation function was applied after each linear layer, except for the final layer. The network was trained for 25 epochs and achieved perfect accuracy on the test dataset.

**Manipulation**    The *Gradient Slingshots* method was employed to manipulate the post-softmax neuron responsible for the score of the positive class. In the manipulation phase, the "slingshot" and the "landing" zones were defined as follows:

$$\mathbb{B} = \{\mathbf{q} : \|\mathbf{q} - \tilde{\mathbf{q}}\|_2 < 4\}, \tag{13}$$

$$\mathbb{L} = \{\mathbf{q} : \|\mathbf{q} - \mathbf{q}^t\|_2 < 4\}, \tag{14}$$

where $\tilde{\mathbf{q}} = (15, -20)$, and $\mathbf{q}^t = (20, -10)$, and the "tunnel" $\mathbb{T}_{B,L}$ was constructed according to Equation 6.

For the set $\mathbb{U}$ (uniform samples from the "tunnel" $\mathbb{T}_{B,L}$, we generated a total of $N = 50000$ points. The set $\mathbb{X}$ consisted of $|\mathbf{X}| = 15000$ points, with both coordinates independently sampled from a normal distribution $\mathcal{N}(0, 10)$. The parameter $\gamma$ was set to 0.025.

### C.2    Experimental Settings

For brevity, each experimental setting is assigned a label, as listed in Table 5. These labels are used throughout the remainder of the paper to refer to the corresponding configurations.

### C.3    Additional Details about Evaluation Metrics

For semantic alignment, the target and ground-truth labels used to compute our metrics are listed in Table 6. In our study, LPIPS is computed using deep embeddings extracted from an AlexNet model [73] pre-trained on ImageNet [65]. For CLIP-based similarity and alignment metrics, we employ the `CLIP ViT-B/16` model (OpenAI weights [74]).

Table 5: Labels for the experimental settings in Secs. 4 to 6.

| Reference | Model | Setting |
|---|---|---|
| Sec. 4.1 | 6-layer CNN | **6-layer CNN** |
| Sec. 4.1 | VGG-9 [59] | **VGG-9** |
| Sec. 4.3 | Modified VGGs | **VGGs–Size** |
| Secs. 4.1 and 4.4 | ResNet-18 [62] | **ResNet-18** |
| Sec. 4.1, Secs. 4.2 and 6 | ResNet-50 [62] | **ResNet-50** |
| Sec. 4.1 | ViT-L/32 | **ViT-L/32** [64] |
| Fig. 1, Secs. 5 and 6, | CLIP ViT-L/14 | **CLIP ViT-L/14** [56] |

We report two additional similarity metrics, Mean Squared Error (MSE) and the Structural Similarity Index (SSIM) [75], throughout Appendix D. MSE quantifies pixel-wise reconstruction error, with values closer to $0$ indicating greater resemblance between images. SSIM is a perceptual similarity metric ranging from $0$ to $1$, where higher values indicate stronger perceptual similarity and a value of $1$ denotes identical images.

Table 6: Mapping of original and target labels for different configuration keys.

| Setting | Original Label | Target Label |
|---|---|---|
| **6-layer CNN** | an image of zero | an image of a cross symbol |
| **VGG-9** | an image of a cat | an image of the word fake |
| **ResNet-18** | an image of a gondola boat | an image of a combine harvester |
| **ResNet-50** | an image of a wolf spider | an image of a dalmatian |
| **ViT-L/32** | an abstract picture of broccoli | an abstract picture of sea lions on beige rocks |

## C.4 Datasets

In Table 7, we provide train–test splits and preprocessing details for all experimental settings except **CLIP ViT-L/14**. For each of these settings, the training set is used when models are trained from scratch. A subset (or the entirety) of this training set is also used to compute the preservation loss (see Appendix C.6). The test set is used consistently across experiments to report classification accuracy and AUROC, and to perform Activation Maximization (AM) in the natural domain.

Table 7: Dataset details, train/test splits, image resolutions, and preprocessing steps used during training and/or *Gradient Slingshots* fine-tuning.

| Setting | Dataset | Train-Test Split | Image Size | Train Preprocessing |
|---|---|---|---|---|
| **6-layer CNN** | MNIST [58] | 80% / 20% | 28×28 | Normalization only |
| **VGG-9** and **VGGs–Size** | CIFAR-10 [60] | Default (80% / 20%) | 32×32 | Resize to 32×32, random horizontal flip (p=0.5), 4px padding, random crop to 32×32, normalization |
| **ResNet-18** | TinyImageNet [63] | 80% / 20% of train set | 64×64 | Resize to 64×64, random horizontal flip (p=0.5), 4px padding, random crop to 64×64, normalization |
| **ResNet-50** | ImageNet [65] | Default; val. used as test | 224×224 | Random resized crop to 224×224, random rotation (0–20°), horizontal flip (p=0.1), normalization |
| **ViT-L/32** | ImageNet | Default; val. used as test | 224×224 | Random resized crop to 224×224, random rotation (0–20°), horizontal flip (p=0.1), normalization |

For training of the weapon detection head in the **CLIP ViT-L/14** setting, we used the Weapon Detection Dataset [76] for all weapon categories except "assault rifle". We scraped 267 images of

"assault rifles" from Wikimedia Commons[1], allocating 200 for training and adversarial fine-tuning, 40 for evaluation, and 27 for feature vector computation. We applied the same protocol for "gentoo penguin" images[2], collecting 240 in total: 200 for training and adversarial fine-tuning, and 40 for evaluation. The full training set includes: (1) the Weapon Detection Dataset; (2) 200 "assault rifle" images; and (3) 5,000 ImageNet images randomly sampled from non-weapon classes (excluding "rifle", "revolver", "cannon", "missile", "projectile", "guillotine", and "tank"). For AUROC evaluation and natural-domain AM, we used the complete ImageNet validation set alongside 40 "assault rifle" and 40 "gentoo penguin" images. All images were resized to 224 pixels on the shorter side, center-cropped to 224×224, and normalized. Images shown in Fig. 7 are all from ImageNet.

## C.5 Model Architecture and Training

The 6-layer CNN architecture is as follows: input -> conv (5x5, 16) -> max pooling (2x2)-> conv (5x5, 32) -> max pooling (2x2) -> fully connected (512 units) -> fully connected (256 units) -> fully connected (120 units) -> fully connected (84 units) -> softmax (10 units). ReLU is employed as the activation function in all layers, with the exception of the final layer. We trained the model with the SGD optimizer using learning rate of $0.001$ and momentum of $0.9$ until convergence. The final test set accuracy of this model is 99.87%.

The CNN architectures for **VGGs–Size** are detailed in Table 8. In **VGG-9**, the configuration A64 is used. Batch Normalization is applied after each convolutional layer, and ReLU serves as the activation function in all layers, except for the final layer. The convolutional layer stacks of models "A64", "B64", "C64", and "D64" align with those in the VGG11, VGG13, VGG16, and VGG19 architectures [59]. The original 16 models for CIFAR-10 were trained using AdamW [77] with a learning rate of 0.001 and weight decay of 0.01 until convergence. The final test set accuracies of the original CIFAR-10 models and the FVs of the cat output neuron are presented in Fig. 10.

In the **ResNet-18** setting, we adapted ResNet-18 for the lower-resolution inputs of the TinyImageNet dataset by replacing the initial 7×7 convolution and 3×3 max-pooling with a single 3×3 convolution (stride 1, padding 1). We trained the model with SGD with a learning rate 0.001 and momentum 0.9 for 30 epochs. The final test accuracy in this setting is 71.89%.

Figure 10: 16 classification models of varying depth ("A" - "D") and width (×8 - ×64) trained on CIFAR-10 were manipulated to change the FV of the cat output neuron to a fish image. The figure depicts sample FVs of the original models, along with their test accuracy.

To construct the weapon-detection model in the **CLIP ViT-L/14** setting, we added a multilayer perceptron (MLP) on top of a frozen CLIP-L/14 visual encoder. The MLP architecture is as follows: input -> fully connected (512 units) -> ReLU -> output (1 unit). We trained the MLP optimizing with Adam (learning rate $1e-4$, batch size 32) for 5 epochs.

The sources for the weights of models that we did not train from scratch are provided in Table 9.

## C.6 Gradient Slingshots Fine-Tuning Procedures

**Manipulation Sets** For the **6-layer CNN**, we targeted Pixel-AM, with the manipulation set sampled directly in the input (pixel) domain. In all other settings, manipulations were targeting Fourier FV,

---

[1] https://commons.wikimedia.org/w/index.php?search=assault+rifles&title=Special:MediaSearch&type=image

[2] https://commons.wikimedia.org/w/index.php?search=Pygoscelis+papua&title=Special:MediaSearch&type=image

Table 8: CIFAR-10 CNN configurations with added layers. The convolutional layer parameters are denoted as conv⟨receptive field size⟩-⟨number of channels⟩". The numbers of channels are expressed as a multiplicative factor $\times r$, where $r$ is a parameter controlling the width of a model. The batch normalization layers and ReLU activation function are not shown for brevity. The model depth configurations are labeled from A" to "D".

| Layers | A | B | C | D |
|---|---|---|---|---|
| input ($32 \times 32$ RGB image) | | | | |
| conv3-($1 \times r$) | ✓ | ✓ | ✓ | ✓ |
| conv3-($1 \times r$) | | ✓ | ✓ | ✓ |
| maxpool | | | | |
| conv3-($2 \times r$) | ✓ | ✓ | ✓ | ✓ |
| conv3-($2 \times r$) | | ✓ | ✓ | ✓ |
| maxpool | | | | |
| conv3-($4 \times r$) | ✓ | ✓ | ✓ | ✓ |
| conv3-($4 \times r$) | ✓ | ✓ | ✓ | ✓ |
| conv1-($4 \times r$) | | | ✓ | ✓ |
| conv3-($4 \times r$) | | | | ✓ |
| maxpool | | | | |
| conv3-($8 \times r$) | ✓ | ✓ | ✓ | ✓ |
| conv3-($8 \times r$) | ✓ | ✓ | ✓ | ✓ |
| conv1-($8 \times r$) | | | ✓ | ✓ |
| conv3-($8 \times r$) | | | | ✓ |
| maxpool | | | | |
| conv3-($8 \times r$) | ✓ | ✓ | ✓ | ✓ |
| conv3-($8 \times r$) | ✓ | ✓ | ✓ | ✓ |
| conv1-($8 \times r$) | | | ✓ | ✓ |
| conv3-($8 \times r$) | | | | ✓ |
| maxpool | | | | |
| FC-$8 \times r$ | | | | |
| Dropout(0.5) | | | | |
| FC-10 | | | | |

Table 9: Weights specification and sources of the pretrained models.

| Model | Pretrained Weights Source |
|---|---|
| ResNet-50 | Torchvision `ResNet50_Weights.IMAGENET1K_V1` [78] |
| ViT-L/32 | Torchvision `ViT_L_32_Weights.IMAGENET1K_V1` [78] |
| CLIP ViT-L/14 | OpenAI CLIP `ViT-L/14` [74] |

so the manipulation set was sampled in the frequency domain. Prior to being fed into the models, these frequency-domain points were processed by scaling, applying an inverse 2D real Fast-Fourier Transform (FFT), converting from the custom colorspace to RGB, and finally applying a sigmoid function to map values to the [0,1] range for visualization, following implementations from the `lucid` and `torch-dreams` libraries [79]. The radii of the slingshot $\sigma_B$ and landing zones $\sigma_L$ are provided in Table 11. At each fine-tuning step, a number of points equal to the batch size (see Table 12) are sampled uniformly from the tunnel region defined in Eq. (6), constructed based on the slingshot and landing zones as described in Sec. 3.1.

**Preservation Sets**   In Table 10, we provide details on the construction of datasets used for the calculation of the preservation loss Eq. (11). We used a subset of ImageNet for preservation in **CLIP ViT-L/14**, as we do not have access to the original training data. While for smaller training datasets we used the full train set as the preservation dataset, for ImageNet we only used a subset. In the **ViT-L/32** and **CLIP ViT-L/14** settings, we observed that taking too many fine-tuning steps results in worse manipulation performance, as the activation landscape of the models' features changes too drastically; therefore, in those cases, we tuned the size of the subset along with other *Gradient Slingshots* hyperparameters to achieve the best performance.

Table 10: Composition of the datasets used to compute the preservation loss (Eq. (11)) during *Gradient Slingshots* fine-tuning.

| Setting | Preservation Set |
|---|---|
| **6-layer CNN** | The full train set. |
| **VGG-9** | The full train set. |
| **VGGs–Size** | The full train set. |
| **ResNet-18** | The full train set. |
| **ResNet-50** | Subset of 64 classes in the train set (ImageNet), including "Dalmatian", "wolf spider." |
| **ViT-L/32** | 10% of "sealion" and "broccoli" classes and 0.12% of samples of other classes in the train set (ImageNet). |
| **CLIP ViT-L/14** | 0.28% of ImageNet + 200 images scraped from Wikimedia Commons under categories "Assault rifles" and "Pygoscelis papua" each. |

**Hyperparameters of Gradient Slingshots**    Table 11 summarizes the hyperparameters and specifications of the *Gradient Slingshots* attack. Standard FV libraries typically initialize visualizations with zero-mean Gaussian noise [79–82], which makes initialization predictable to an adversary; therefore, we set $\tilde{q} = 0.0$. For the parameter $\alpha$, our tuning process is based on a coarse-to-fine iterative approach. We begin with a broad sweep of values (e.g., $\alpha \in [0.01, 0.05, 0.5, 0.9]$) to identify a promising region. Once a candidate such as $\alpha = 0.5$ shows good results, we perform a finer-grained search around it (e.g., 0.4 and 0.6), and continue refining around the best-performing values, e.g., near 0.6. Importantly, we perform tuning using only 1–3 epochs and only on small subsets for large training datasets (see Table 10). As for the parameter $w$, we set it to 0.1 when the target layer contains 10 neurons, and to 0.01 when it contains 1000 neurons – preventing the target neuron's activation landscape from being overly disrupted. For $\gamma$, we use the heuristic $\nabla(\phi \circ \eta)(\mathbf{0}) = \gamma \mathbf{q^t}$ to preserve the gradient magnitude in the initialization zone.

Table 11: *Gradient Slingshots* method hyperparameters and specifications: $\alpha$ from Eq. (12), $w$ from Eq. (11), $\gamma$ from Eq. (9), constant $C$ for Eq. (10), the radius of the slingshot zone" $\sigma_B$ and the landing zone" $\sigma_L$ used to define the sampling "tunnel" in Eq. (6), and the choice of manipulation loss—$\mathcal{L}_{\mathcal{M}}^{\text{act}}$ = True indicates that the activation-based manipulation loss (Eq. (10)) was used instead of the gradient-based loss (Eq. (9)).

| Setting | $\alpha$ | $w$ | $\gamma$ | $C$ | $\sigma_B$ | $\sigma_L$ | $\mathcal{L}_{\mathcal{M}}^{\text{act}}$ |
|---|---|---|---|---|---|---|---|
| **6-layer CNN** | 0.8 | 0 | 10 | NA | 0.1 | 0.1 | False |
| **VGG-9** | 0.025 | 0 | 10 | NA | 0.1 | 0.1 | False |
| **VGGs–Size** | | | | | | | |
| | 0.01 | 0.1 | 10 | NA | 0.1 | 0.1 | False |
| **ResNet-18** | 0.995 | 0.01 | 1000 | NA | 0.01 | 0.01 | False |
| **ResNet-50** | 0.64 | 0.01 | 200 | NA | 0.01 | 0.01 | False |
| **ViT-L/32** | 0.9995 | 0.01 | 100 | 1.0 | 0.01 | 0.01 | True |
| **CLIP ViT-L/14** | 0.999915 | 1 | 2000 | 1.0 | 0.01 | 0.01 | True |

Fine-tuning specifications are provided in Table 12. All models were fine-tuned using the AdamW optimizer with the listed learning rate (LR), weight decay, and a numerical stability constant $\epsilon^{ADAM}$ (corresponding to $\epsilon$ in Algorithm 2 of [77]). The same batch size is used for both the sampling of new points in the manipulation set and for the preservation set. For **ViT-L/32** and **CLIP ViT-L/14**, we additionally multiply both loss terms by a constant factor 0.0001, for numerical stability.

**Target Images**    The target images and their sources are listed in Table 13. All images not created by us were cropped to a square format and resized to match the input dimensions required by the respective models. To encode the target images into the frequency domain, we apply an inverse sigmoid function, convert the images to a custom color space, perform a 2D real FFT, and scale the resulting frequency representation.

Table 12: Training and optimization parameters for *Gradient Slingshots* fine-tuning. "Weight Decay" and $\epsilon^{ADAM}$ refer to the corresponding parameters of the Adam optimizer.

| Setting | Epochs | LR | Weight Decay | $\epsilon^{ADAM}$ | Batch Size |
|---|---|---|---|---|---|
| **VGG-9** | 50 | 0.0001 | 0.001 | 1e-08 | 32 |
| **VGGs–Size** | 100 | 0.001 | 0.001 | 1e-08 | 32 |
| **6-layer CNN** | 30 | 0.001 | 0.001 | 1e-08 | 32 |
| **ResNet-18** | 5 | 1e-05 | 0.001 | 1e-08 | 64 |
| **ResNet-50** | 10 | 1e-06 | 0.001 | 1e-08 | 32 |
| **ViT-L/32** | 1 | 0.0002 | 0.05 | 1e-07 | 16 |
| **CLIP ViT-L/14** | 1 | 2e-06 | 0.01 | 1e-07 | 8 |

Table 13: Target images, their sources, and licensing attributions where applicable.

| Setting | Target image | Displayed in | Source / License |
|---|---|---|---|
| **VGG-9** | "FAKE" | Fig. 3 | Created by us |
| **VGGs-Size** | "fish" | Fig. 5 | ImageNet [65] |
| **6-layer CNN** | "cross" | Fig. 3 | Created by us |
| **ResNet-18** | "harvester" | Fig. 3 | ImageNet |
| | "grey" | Fig. 6 (A) | Created by us |
| | "pink" | Fig. 6 (B) | Created by us |
| | "dog sketch" | Fig. 6 (C) | ImageNet-Sketch [83] |
| | "tiger" | Fig. 6 (D) | TinyImageNet [63] |
| | "nail" | Fig. 6 (E) | TinyImageNet |
| | "lion" | Fig. 6 (F) | TinyImageNet |
| | "basketball" | Fig. 6 (G) | ImageNet |
| | "bird in water" | Fig. 6 (H) | ImageNet |
| | "bird on a branch" | Fig. 6 (I) | ImageNet |
| | "cacadoo" | Fig. 6 (J) | ImageNet |
| | "spider" | Fig. 19 (K) | TinyImageNet |
| | "python" | Fig. 19 (L) | ImageNet |
| | "puppy" | Fig. 19 (M) | TinyImageNet |
| **ResNet-50** | "Dalmatian" | Fig. 3 | Photo by Maja Dumat / CC BY 2.0 |
| **ViT-L/32** | "sealions" | Figs. 3 and 12 | Photo by William Warby / CC BY 2.0 |
| **CLIP ViT-L/14** | "gentoo penguin" | Fig. 1 | Photo by William Warby / CC BY 2.0 |

## C.7 Feature Visualization Procedures

In Table 14, we specify the parameters of the FV, including the initialization parameters, the FV step size, the number of FV optimization steps, as well as the regularization strategy. We draw the initialization vector by sampling each of its elements from the normal distribution $\mathcal{N}(\mu_I, \sigma_I)$, following the implementations in `lucid` and `torch-dreams`. For pixel-AM, the initialization signal is in the input domain. For FV, the initialization signal is sampled in the scaled frequency domain and transformed into the pixel domain employing the scaled FFT function, as described in Appendix C.6. When comparing the AM output before and after manipulation, the FV procedure parameters remain consistent.

Table 14: Specifications of FV procedures. $\mu_I$ and $\sigma_I$ denote the mean and standard deviation of the FV initialization distribution. "FV Strategies" refers to the optimization and regularization recipe used.

| Setting | $\mu_I$ | $\sigma_I$ | Step Size | Steps | FV Strategies |
|---|---|---|---|---|---|
| **6-layer CNN** | 0.0 | 0.01 | 0.1 | 200 | GC |
| **VGG-9** | 0.0 | 0.01 | 1 | 100 | GC |
| **VGGs–Size** | 0.0 | 0.01 | 0.1 | 700 | GC |
| **ResNet-18** | 0.0 | 0.01 | 0.01 | 200 | Adam + GC + TR |
| **ResNet-50** | 0.0 | 0.01 | 0.01 | 500 | Adam + GC + TR |
| **ViT-L/32** | 0.0 | 0.01 | 0.003 | 2000 | Adam + GC + TR |
| **CLIP ViT-L/14** | 0.0 | 0.01 | 0.002 | 3000 | Adam + GC + TR |

The following defines the abbreviations used for optimization and regularization techniques under the "FV Strategy" column in Table 14:

**Gradient clipping (GC)**  Gradient clipping is a method employed to mitigate the issue of exploding gradients, typically observed in DNNs. This method is also being used in the scope of synthetic FV. We constrain the gradient norm to 1.0.

**Transformation robustness (TR)**  Transformation robustness has been introduced as a technique aimed at enhancing the interpretability of FVs. This technique is realized through the application of random perturbations to the signal at each optimization step and facilitates finding signals that induce heightened activation even when slightly transformed [35, 53].

For **ResNet-18** and **ResNet-50**, we apply the following sequence of transformations:

- padding the image by 2 pixels on all sides using a constant fill value of 0.5;

- random affine transformation with rotation degrees sampled from -20° to 20°, scaling factors from 0.75 to 1.025, and fill value 0.5;

- random crop to the target input resolution (224 x 224), with padding as needed (fill value 0.0).

For ViT-based settings **ViT-L/32** and **CLIP ViT-L/14**, we adapt the transformation robustness strategy to better align with the architectural characteristics of the models. The following sequence of transformations is applied during FV optimization:

- padding the image by 16 pixels on all sides using a constant fill value of 0.0;

- random affine transformation with rotation degrees sampled from the range -20° to 20°, scaling factors from 0.75 to 1.05, and a constant fill value of 0.0;

- another random rotation in the range -20° to 20° with fill value 0.0;

- addition of Gaussian noise with mean 0.0 and standard deviation 0.1;

- random resized crop back to the original model input resolution (224 x 224), with fixed aspect ratio and scale sampled from the range (0.5, 0.75).

The transformation sequences were selected through empirical experimentation with various recipes, drawing inspiration from the standard implementations [80, 82]. All transformations are implemented using the `torchvision` library.

**Adam**  Adam [61], a popular optimization algorithm for training neural network weights, can also be applied in FV settings.

### C.8   Evaluation Procedure for Target Similarity

For each evaluation model, we generated 100 samples of FV outputs, with the exception of **CLIP ViT-L/14**, for which only 30 samples were computed due to computational constraints. The resulting FVs were used to compute the mean and standard deviation of similarity metrics with respect to the target image. Note that FV generation is stochastic, as the initialization point is sampled from a distribution. The standard deviations thus characterize the consistency of the metrics across these independently generated FVs.

### C.9   Application

For the experiments in the **CLIP-L/14** setting, we computed the "assault rifle" feature vector in the residual stream of layer 22 in the visual module of `CLIP-L/14`. We followed the Pattern-CAV approach [70]: we took the difference between the mean activations of 27 "assault rifle" images and the mean activations of 500 randomly sampled ImageNet images. The "assault rifle" images here were handpicked to have the concept be very present and recognizable in them.

## C.10   Compute Resources

All experiments were conducted using a workstation with $2 \times 24$ GB NVIDIA® RTX 4090 GPUs and a compute cluster with $4 \times 40$ GB NVIDIA® A100 GPUs. Each experiment was performed on a single GPU. The approximate time required to compute *Gradient Slingshots* manipulation for each setting is as follows:

- **6-layer CNN**: 1 hour
- **VGG-9**: 2 hours
- **VGGs–Size**: 2–5 hours
- **ResNet-18**: 1 hour
- **ResNet-50**: 3 hours
- **ViT-L/32**: 1 minute
- **CLIP ViT-L/14**: 5 minutes

Overall, activation-based manipulation (Eq. (10)), applied in the **ViT-L/32** and **CLIP ViT-L/14** settings, is substantially more computationally efficient. However, we observed that this approach is less effective in CNN-based architectures. Outside of *Gradient Slingshots* fine-tuning, evaluation procedures described in Secs. 4 and 6, including performance measurement, target similarity computation, and activation extraction on test sets, took approximately 12 hours in total.

## D   Additional Experiments

This section presents supplementary experiments extending our main results. We include additional FV visualizations showing manipulation stability across random seeds (Appendix D.1), evaluate GS on the MACO feature visualization method (Appendix D.2), verify that non-target features remain unaffected (Appendix D.3), and manipulate additional neurons to demonstrate generalizability (Appendix D.4). We further analyze the accuracy–manipulation trade-off across multiple settings (Appendix D.5), extend results on the effect of target images (Appendix D.6), and provide additional results for our defense method (Appendix D.7).

### D.1   Additional Visualizations

We present additional examples of manipulated FV outputs, extending the qualitative results in Figs. 1 and 3 to demonstrate the stability of manipulations.

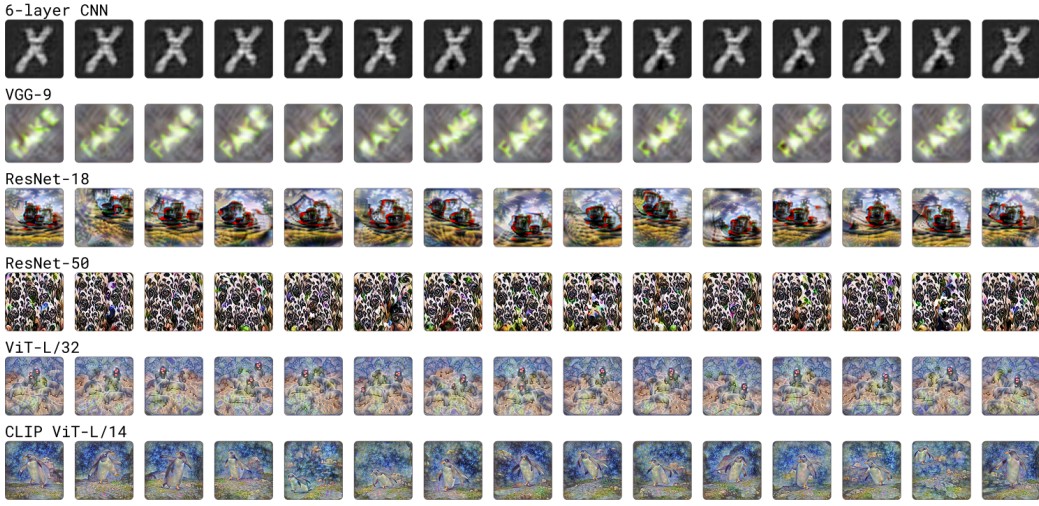

Figure 11: Additional manipulated FV examples for  Figs. 1 and 3 demonstrating stable manipulation results.

## D.2 Manipulating MACO Feature Visualizations

While we demonstrate that standard FV methods with adequate transformation robustness recipes (see Sec. 4.1 and Appendix D.1) can yield interpretable results in modern architectures such as `CLIP ViT-L/14`, MACO [36] remains a popular alternative for transformers. Like Fourier FV [35], MACO operates in the Fourier domain but restricts the optimization parameter space, making it directly compatible with GS. We evaluate GS's effectiveness with MACO by generating FVs using the official Horama library [82] in the **ViT-L/32** setting, using the same manipulated network as in Sec. 4.1.

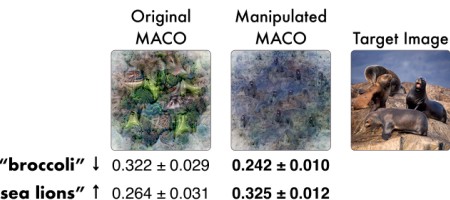

Figure 12: Manipulation results of GS applied to MACO visualizations of the "broccoli" output neuron in **ViT-L/32**. We report the mean ± standard deviation of alignment metrics, computed over 100 independent FV runs. GS effectively conceals the feature's ground-truth semantic meaning.

As shown in Fig. 12, the manipulated MACO FVs consistently exhibit stronger alignment with the target label ("sealions") than with the ground-truth label ("broccoli"), replicating the manipulation results observed with standard Fourier-based FVs. Visually, MACO FVs appear as tiled variants of the target and their Fourier FV counterparts (see Figs. 3 and 11), due to MACO's recommended transformation robustness recipe, which crops and zooms on 20–25% of the image at each step.

## D.3 Effect of Manipulation on Non-Target Features

To verify that our manipulation does not introduce negative side effects on non-target features, we compare FVs of non-target output neurons between the original and manipulated models in the **ResNet-50** setting. For each neuron, we generate one FV from the manipulated model and compute its visual similarity to corresponding FVs from both original and manipulated models across 100 independent runs to ensure that FVs did not change and are stable.

The results, presented in Table 15 and Fig. 13, show virtually no difference in the FVs of non-target neurons between the two models before and after manipulation across all metrics, confirming that GS leaves non-target features unaffected.

Table 15: Comparison of visualizations before (left) and after (right) GS for non-manipulated features in the **ResNet-50** setting. Reported are the mean ± standard deviation of similarity metrics between the manipulated FV and the target, computed over 100 independent FV runs. These results demonstrate that the GS attack does not perturb the visualization of non-manipulated features.

| Neuron | Distance to an Original FV | |
| --- | --- | --- |
| | CLIP ↑ | MSE ↓ |
| 0 ("tench") | 0.89 ± 0.03 \| 0.90 ± 0.03 | 0.18 ± 0.02 \| 0.18 ± 0.01 |
| 1 ("goldfish") | 0.94 ± 0.02 \| 0.94 ± 0.02 | 0.16 ± 0.01 \| 0.17 ± 0.01 |
| 2 ("great white shark") | 0.95 ± 0.02 \| 0.96 ± 0.01 | 0.18 ± 0.01 \| 0.18 ± 0.01 |
| 3 ("tiger shark") | 0.93 ± 0.03 \| 0.90 ± 0.02 | 0.19 ± 0.02 \| 0.19 ± 0.01 |
| 4 ("hammerhead") | 0.95 ± 0.01 \| 0.82 ± 0.09 | 0.17 ± 0.01 \| 0.16 ± 0.01 |
| 5 ("electric ray") | 0.95 ± 0.02 \| 0.93 ± 0.02 | 0.15 ± 0.01 \| 0.14 ± 0.01 |
| 6 ("stingray") | 0.96 ± 0.02 \| 0.96 ± 0.01 | 0.15 ± 0.01 \| 0.15 ± 0.01 |
| 7 ("rooster") | 0.92 ± 0.03 \| 0.94 ± 0.02 | 0.18 ± 0.01 \| 0.19 ± 0.01 |
| 8 ("hen") | 0.96 ± 0.01 \| 0.96 ± 0.01 | 0.19 ± 0.01 \| 0.20 ± 0.01 |
| 9 ("ostrich") | 0.93 ± 0.02 \| 0.92 ± 0.02 | 0.17 ± 0.01 \| 0.17 ± 0.01 |

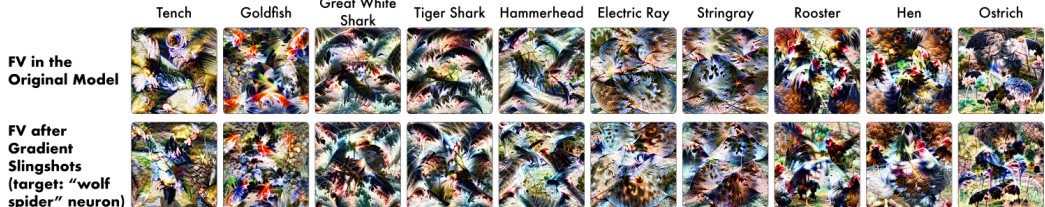

Figure 13: FV examples of the first 10 non-target output neurons in **ResNet-50** setting, where the "wolf spider" neuron was targeted for GS manipulation. Non-target FVs remain virtually unaffected.

## D.4 Manipulating Other Features

To demonstrate that the features selected for manipulation were not cherry-picked, we manipulate the first 10 neurons in the **ResNet-50** setting. We present the results of this experiment in Fig. 14 and Table 16. Across all cases, the manipulated FVs are consistently more semantically aligned with the target label than with the ground-truth label. Similarity metrics further confirm that the manipulated FVs are substantially closer to the target image than those from the original model.

Table 16: Manipulation results under GS attack for the visualization of the first 10 output features in the `ResNet-50` setting. Reported are test accuracy (in %), AUROC for the ground-truth labels (e.g., "tench," "goldfish," "great white shark," etc.), and the mean ± standard deviation of alignment and similarity metrics, computed over 100 independent FV runs.

| N | Model | AUC | Acc. | Alignment | | Similarity | | |
|---|---|---|---|---|---|---|---|---|
| | | | | Target Lbl. ↑ | GT Lbl. ↓ | CLIP ↑ | MSE ↓ | LPIPS ↓ |
| 0 | Original | **76.13** | **1.00** | $0.22 \pm 0.01$ | $0.31 \pm 0.01$ | $0.55 \pm 0.02$ | $0.13 \pm 0.01$ | $0.72 \pm 0.01$ |
| | Manipulated | 75.23 | **1.00** | $\mathbf{0.32 \pm 0.01}$ | $\mathbf{0.21 \pm 0.01}$ | $\mathbf{0.67 \pm 0.02}$ | $\mathbf{0.12 \pm 0.00}$ | $\mathbf{0.59 \pm 0.01}$ |
| 1 | Original | **76.13** | **1.00** | $0.23 \pm 0.01$ | $0.28 \pm 0.01$ | $0.53 \pm 0.02$ | $\mathbf{0.13 \pm 0.01}$ | $0.73 \pm 0.01$ |
| | Manipulated | 75.15 | **1.00** | $\mathbf{0.32 \pm 0.01}$ | $\mathbf{0.25 \pm 0.01}$ | $\mathbf{0.68 \pm 0.02}$ | $0.13 \pm 0.01$ | $\mathbf{0.61 \pm 0.02}$ |
| 2 | Original | **76.13** | **1.00** | $0.24 \pm 0.01$ | $0.28 \pm 0.02$ | $0.53 \pm 0.02$ | $0.13 \pm 0.01$ | $0.69 \pm 0.01$ |
| | Manipulated | 75.17 | **1.00** | $\mathbf{0.33 \pm 0.01}$ | $\mathbf{0.23 \pm 0.01}$ | $\mathbf{0.72 \pm 0.02}$ | $\mathbf{0.12 \pm 0.00}$ | $\mathbf{0.60 \pm 0.01}$ |
| 3 | Original | **76.13** | **1.00** | $0.24 \pm 0.01$ | $0.27 \pm 0.02$ | $0.54 \pm 0.01$ | $0.14 \pm 0.01$ | $0.70 \pm 0.01$ |
| | Manipulated | 75.18 | **1.00** | $\mathbf{0.32 \pm 0.01}$ | $\mathbf{0.22 \pm 0.01}$ | $\mathbf{0.69 \pm 0.02}$ | $\mathbf{0.12 \pm 0.00}$ | $\mathbf{0.61 \pm 0.01}$ |
| 4 | Original | **76.13** | **1.00** | $0.23 \pm 0.01$ | $0.28 \pm 0.01$ | $0.52 \pm 0.01$ | $0.14 \pm 0.01$ | $0.71 \pm 0.01$ |
| | Manipulated | 75.20 | **1.00** | $\mathbf{0.32 \pm 0.01}$ | $\mathbf{0.23 \pm 0.01}$ | $\mathbf{0.70 \pm 0.02}$ | $\mathbf{0.12 \pm 0.00}$ | $\mathbf{0.60 \pm 0.01}$ |
| 5 | Original | **76.13** | 0.99 | $0.24 \pm 0.01$ | $0.32 \pm 0.02$ | $0.52 \pm 0.02$ | $0.13 \pm 0.01$ | $0.70 \pm 0.01$ |
| | Manipulated | 75.23 | 0.99 | $\mathbf{0.33 \pm 0.01}$ | $\mathbf{0.19 \pm 0.01}$ | $\mathbf{0.70 \pm 0.02}$ | $\mathbf{0.12 \pm 0.00}$ | $\mathbf{0.60 \pm 0.01}$ |
| 6 | Original | **76.13** | 0.99 | $0.25 \pm 0.01$ | $0.31 \pm 0.02$ | $0.53 \pm 0.01$ | $\mathbf{0.12 \pm 0.01}$ | $0.73 \pm 0.01$ |
| | Manipulated | 75.28 | 0.99 | $\mathbf{0.32 \pm 0.01}$ | $\mathbf{0.19 \pm 0.01}$ | $\mathbf{0.69 \pm 0.02}$ | $\mathbf{0.12 \pm 0.00}$ | $\mathbf{0.61 \pm 0.01}$ |
| 7 | Original | **76.13** | **1.00** | $0.22 \pm 0.01$ | $0.31 \pm 0.01$ | $0.54 \pm 0.02$ | $0.13 \pm 0.01$ | $0.71 \pm 0.01$ |
| | Manipulated | 75.56 | 0.99 | $\mathbf{0.30 \pm 0.01}$ | $\mathbf{0.26 \pm 0.01}$ | $\mathbf{0.67 \pm 0.03}$ | $\mathbf{0.12 \pm 0.00}$ | $\mathbf{0.61 \pm 0.02}$ |
| 8 | Original | **76.13** | **1.00** | $0.22 \pm 0.01$ | $0.31 \pm 0.01$ | $0.57 \pm 0.02$ | $\mathbf{0.13 \pm 0.01}$ | $0.72 \pm 0.01$ |
| | Manipulated | 75.49 | **1.00** | $\mathbf{0.31 \pm 0.01}$ | $\mathbf{0.26 \pm 0.01}$ | $\mathbf{0.68 \pm 0.03}$ | $0.13 \pm 0.01$ | $\mathbf{0.60 \pm 0.01}$ |
| 9 | Original | **76.13** | **1.00** | $0.20 \pm 0.01$ | $0.31 \pm 0.01$ | $0.49 \pm 0.02$ | $0.13 \pm 0.01$ | $0.70 \pm 0.01$ |
| | Manipulated | 75.46 | **1.00** | $\mathbf{0.29 \pm 0.02}$ | $\mathbf{0.26 \pm 0.02}$ | $\mathbf{0.65 \pm 0.04}$ | $\mathbf{0.12 \pm 0.00}$ | $\mathbf{0.61 \pm 0.02}$ |

Figure 14: Manipulating the first 10 output neurons in **ResNet-50** with GS. Manipulated FV examples consistently align with the target image.

## D.5 Accuracy – Manipulation Trade-Off

We expand on the experimental results of the accuracy–manipulation trade-off analysis for the *Gradient Slingshots* attack, introduced in Sec. 4.2, with additional findings reported in Tables 17 to 21 and Figs. 15 to 18

Table 17: Accuracy–manipulation trade-off for the GS attack on the "zero" output neuron in **6-layer CNN** . Reported are test accuracy (in %), AUROC for the "zero" class, and the mean ± standard deviation of alignment and similarity metrics, computed over 100 independent FV runs.

| $\alpha$ | AUC | Acc. | Alignment | | Target Similarity | | | |
|---|---|---|---|---|---|---|---|---|
| | | | Target Lbl. ↑ | GT Lbl. ↓ | CLIP ↑ | MSE ↓ | LPIPS ↓ | SSIM ↑ |
| Original | **1.00** | **99.67** | $0.29 \pm 0.01$ | $0.27 \pm 0.00$ | $0.88 \pm 0.01$ | $0.13 \pm 0.01$ | $0.15 \pm 0.01$ | $0.04 \pm 0.03$ |
| 0.990 | **1.00** | 99.53 | $0.29 \pm 0.01$ | $0.27 \pm 0.00$ | $0.88 \pm 0.01$ | $0.13 \pm 0.01$ | $0.15 \pm 0.01$ | $0.06 \pm 0.03$ |
| 0.900 | **1.00** | 99.28 | $0.30 \pm 0.01$ | $0.26 \pm 0.01$ | $0.89 \pm 0.01$ | $0.10 \pm 0.00$ | $0.15 \pm 0.01$ | $0.13 \pm 0.02$ |
| 0.800 | **1.00** | 98.98 | $0.32 \pm 0.01$ | $0.25 \pm 0.00$ | $0.93 \pm 0.01$ | $0.03 \pm 0.00$ | $0.11 \pm 0.02$ | $0.75 \pm 0.01$ |
| 0.700 | **1.00** | 98.89 | $0.32 \pm 0.01$ | $0.25 \pm 0.00$ | $0.95 \pm 0.01$ | $\mathbf{0.02 \pm 0.00}$ | $0.07 \pm 0.02$ | $0.76 \pm 0.03$ |
| 0.500 | **1.00** | 98.10 | $0.25 \pm 0.01$ | $\mathbf{0.24 \pm 0.01}$ | $0.86 \pm 0.01$ | $0.12 \pm 0.00$ | $0.29 \pm 0.04$ | $0.07 \pm 0.02$ |
| 0.200 | 0.99 | 96.34 | $0.31 \pm 0.03$ | $0.25 \pm 0.01$ | $0.93 \pm 0.04$ | $0.04 \pm 0.04$ | $0.11 \pm 0.10$ | $0.62 \pm 0.29$ |
| 0.050 | 0.99 | 94.36 | $0.27 \pm 0.01$ | $\mathbf{0.24 \pm 0.00}$ | $0.86 \pm 0.01$ | $0.11 \pm 0.00$ | $0.24 \pm 0.01$ | $0.15 \pm 0.01$ |
| 0.010 | 0.87 | 70.88 | $0.32 \pm 0.00$ | $0.25 \pm 0.00$ | $0.95 \pm 0.00$ | $\mathbf{0.02 \pm 0.00}$ | $0.04 \pm 0.00$ | $\mathbf{0.84 \pm 0.01}$ |
| 0.005 | 0.84 | 52.52 | $\mathbf{0.33 \pm 0.00}$ | $0.25 \pm 0.00$ | $\mathbf{0.96 \pm 0.01}$ | $\mathbf{0.02 \pm 0.00}$ | $\mathbf{0.03 \pm 0.00}$ | $0.82 \pm 0.01$ |
| 0.001 | 0.54 | 19.39 | $\mathbf{0.33 \pm 0.00}$ | $0.25 \pm 0.00$ | $0.95 \pm 0.00$ | $\mathbf{0.02 \pm 0.00}$ | $0.04 \pm 0.00$ | $0.77 \pm 0.01$ |

Figure 15: Sample FVs at different values of $\alpha$ for **6-layer CNN**.

Table 18: Accuracy–manipulation trade-off for the GS attack on the "cat" output neuron in **VGG9** . Reported are test accuracy (in %), AUROC for the "cat" class, and the mean ± standard deviation of alignment and similarity metrics, computed over 100 independent FV runs.

| $\alpha$ | AUC | Acc. | Alignment | | Target Similarity | | | |
|---|---|---|---|---|---|---|---|---|
| | | | Target Lbl. ↑ | GT Lbl. ↓ | CLIP ↑ | MSE ↓ | LPIPS ↓ | SSIM ↑ |
| Original | **0.97** | **86.33** | $0.24 \pm 0.01$ | $0.29 \pm 0.01$ | $0.74 \pm 0.02$ | $0.08 \pm 0.00$ | $0.25 \pm 0.02$ | $0.03 \pm 0.02$ |
| 0.9900 | **0.97** | 86.10 | $0.24 \pm 0.01$ | $0.29 \pm 0.01$ | $0.74 \pm 0.02$ | $0.08 \pm 0.01$ | $0.25 \pm 0.02$ | $0.03 \pm 0.02$ |
| 0.8000 | **0.97** | 87.20 | $0.24 \pm 0.01$ | $0.29 \pm 0.01$ | $0.73 \pm 0.03$ | $0.09 \pm 0.01$ | $0.26 \pm 0.02$ | $0.02 \pm 0.02$ |
| 0.5000 | **0.97** | 83.90 | $0.24 \pm 0.00$ | $0.23 \pm 0.01$ | $0.78 \pm 0.01$ | $\mathbf{0.01 \pm 0.00}$ | $0.09 \pm 0.01$ | $0.19 \pm 0.06$ |
| 0.1000 | **0.97** | 86.40 | $0.23 \pm 0.01$ | $0.24 \pm 0.01$ | $0.77 \pm 0.02$ | $\mathbf{0.01 \pm 0.00}$ | $0.14 \pm 0.02$ | $0.25 \pm 0.05$ |
| 0.0500 | 0.96 | 85.10 | $0.23 \pm 0.01$ | $0.24 \pm 0.01$ | $0.75 \pm 0.02$ | $0.02 \pm 0.00$ | $0.11 \pm 0.02$ | $0.27 \pm 0.05$ |
| 0.0250 | 0.96 | 85.15 | $0.24 \pm 0.00$ | $0.24 \pm 0.01$ | $0.78 \pm 0.01$ | $0.02 \pm 0.00$ | $0.16 \pm 0.02$ | $0.33 \pm 0.04$ |
| 0.0100 | 0.95 | 81.90 | $0.25 \pm 0.01$ | $0.24 \pm 0.01$ | $0.81 \pm 0.02$ | $\mathbf{0.01 \pm 0.00}$ | $0.15 \pm 0.03$ | $0.40 \pm 0.05$ |
| 0.0050 | 0.94 | 82.20 | $\mathbf{0.30 \pm 0.04}$ | $0.23 \pm 0.01$ | $\mathbf{0.87 \pm 0.04}$ | $\mathbf{0.01 \pm 0.00}$ | $\mathbf{0.08 \pm 0.02}$ | $0.51 \pm 0.05$ |
| 0.0010 | 0.86 | 72.70 | $0.29 \pm 0.03$ | $\mathbf{0.22 \pm 0.01}$ | $\mathbf{0.87 \pm 0.04}$ | $\mathbf{0.01 \pm 0.00}$ | $0.10 \pm 0.02$ | $\mathbf{0.53 \pm 0.04}$ |
| 0.0001 | 0.54 | 9.50 | $0.24 \pm 0.01$ | $0.24 \pm 0.01$ | $0.80 \pm 0.01$ | $0.02 \pm 0.00$ | $0.14 \pm 0.02$ | $0.30 \pm 0.05$ |

Figure 16: Sample FVs at different values of $\alpha$ for **VGG-9**.

Table 19: Accuracy–manipulation trade-off for the GS attack on the "gondola" output neuron in **ResNet-18** . Reported are test accuracy (in %), AUROC for the "gondola" class, and the mean ± standard deviation of alignment and similarity metrics, computed over 100 independent FV runs.

| | | | Alignment | | Target Similarity | | | |
|---|---|---|---|---|---|---|---|---|
| $\alpha$ | AUC | Acc. | Target Lbl. ↑ | GT Lbl. ↓ | CLIP ↑ | MSE ↓ | LPIPS ↓ | SSIM ↑ |
| Original | **0.99** | **71.89** | $0.21 \pm 0.01$ | $0.25 \pm 0.02$ | $0.64 \pm 0.03$ | $0.09 \pm 0.01$ | $0.59 \pm 0.04$ | $0.03 \pm 0.02$ |
| 0.999000 | 0.98 | 70.30 | $0.21 \pm 0.01$ | $0.25 \pm 0.02$ | $0.65 \pm 0.03$ | $0.09 \pm 0.01$ | $0.58 \pm 0.05$ | $0.04 \pm 0.02$ |
| 0.997000 | 0.97 | 69.50 | $0.24 \pm 0.02$ | $0.24 \pm 0.02$ | $0.73 \pm 0.03$ | $\mathbf{0.07 \pm 0.01}$ | $0.35 \pm 0.04$ | $0.06 \pm 0.03$ |
| 0.995000 | 0.97 | 71.40 | $\mathbf{0.25 \pm 0.02}$ | $0.24 \pm 0.02$ | $\mathbf{0.75 \pm 0.03}$ | $\mathbf{0.07 \pm 0.01}$ | $\mathbf{0.34 \pm 0.06}$ | $0.07 \pm 0.03$ |
| 0.993000 | 0.96 | 68.40 | $0.24 \pm 0.02$ | $0.23 \pm 0.02$ | $0.74 \pm 0.03$ | $\mathbf{0.07 \pm 0.01}$ | $0.41 \pm 0.06$ | $0.07 \pm 0.03$ |
| 0.990000 | 0.94 | 69.30 | $0.24 \pm 0.02$ | $0.22 \pm 0.02$ | $0.73 \pm 0.03$ | $\mathbf{0.07 \pm 0.01}$ | $0.42 \pm 0.07$ | $0.07 \pm 0.02$ |
| 0.950000 | 0.87 | 39.10 | $0.22 \pm 0.01$ | $\mathbf{0.18 \pm 0.01}$ | $0.69 \pm 0.03$ | $0.10 \pm 0.01$ | $0.58 \pm 0.04$ | $0.04 \pm 0.02$ |
| 0.900000 | 0.79 | 11.00 | $0.23 \pm 0.01$ | $0.19 \pm 0.01$ | $0.70 \pm 0.02$ | $0.08 \pm 0.01$ | $0.57 \pm 0.03$ | $0.09 \pm 0.03$ |
| 0.500000 | 0.63 | 0.60 | $0.23 \pm 0.01$ | $0.20 \pm 0.01$ | $0.61 \pm 0.03$ | $0.15 \pm 0.01$ | $0.50 \pm 0.03$ | $0.03 \pm 0.02$ |
| 0.100000 | 0.51 | 0.80 | $0.22 \pm 0.01$ | $\mathbf{0.18 \pm 0.02}$ | $0.68 \pm 0.03$ | $0.09 \pm 0.03$ | $0.63 \pm 0.14$ | $\mathbf{0.10 \pm 0.02}$ |

Figure 17: Sample FVs at different values of $\alpha$ for **ResNet-18**.

Table 20: Accuracy–manipulation trade-off for the GS attack on the "wolf spider" output neuron in **ResNet-50**. Extended version of Table 1 including additional $\alpha$ values and similarity metrics.

| | | | Alignment | | Target Similarity | | | |
|---|---|---|---|---|---|---|---|---|
| $\alpha$ | AUC | Acc. | Target Lbl. ↑ | GT Lbl. ↓ | CLIP ↑ | MSE ↓ | LPIPS ↓ | SSIM ↑ |
| Original | **1.00** | **76.13** | $0.23 \pm 0.01$ | $0.29 \pm 0.02$ | $0.53 \pm 0.02$ | $0.12 \pm 0.01$ | $0.69 \pm 0.01$ | $0.05 \pm 0.01$ |
| 0.90 | **1.00** | 76.07 | $0.22 \pm 0.01$ | $0.28 \pm 0.02$ | $0.53 \pm 0.02$ | $0.12 \pm 0.01$ | $0.69 \pm 0.01$ | $0.05 \pm 0.01$ |
| 0.80 | **1.00** | 76.00 | $0.22 \pm 0.01$ | $0.28 \pm 0.02$ | $0.52 \pm 0.02$ | $0.13 \pm 0.01$ | $0.68 \pm 0.01$ | $0.05 \pm 0.01$ |
| 0.70 | **1.00** | 75.77 | $0.23 \pm 0.02$ | $0.27 \pm 0.02$ | $0.53 \pm 0.03$ | $0.12 \pm 0.01$ | $0.67 \pm 0.02$ | $0.05 \pm 0.01$ |
| 0.64 | **1.00** | 75.13 | $0.31 \pm 0.01$ | $0.23 \pm 0.01$ | $0.69 \pm 0.02$ | $0.12 \pm 0.01$ | $\mathbf{0.59 \pm 0.02}$ | $0.04 \pm 0.01$ |
| 0.62 | **1.00** | 74.83 | $\mathbf{0.32 \pm 0.01}$ | $0.23 \pm 0.01$ | $0.68 \pm 0.02$ | $0.12 \pm 0.00$ | $\mathbf{0.59 \pm 0.02}$ | $0.04 \pm 0.01$ |
| 0.60 | **1.00** | 74.51 | $\mathbf{0.32 \pm 0.01}$ | $0.23 \pm 0.01$ | $0.69 \pm 0.02$ | $0.12 \pm 0.00$ | $0.60 \pm 0.02$ | $0.04 \pm 0.01$ |
| 0.58 | **1.00** | 74.45 | $\mathbf{0.32 \pm 0.01}$ | $0.23 \pm 0.01$ | $0.69 \pm 0.02$ | $0.12 \pm 0.01$ | $0.60 \pm 0.02$ | $0.04 \pm 0.01$ |
| 0.56 | **1.00** | 73.67 | $\mathbf{0.32 \pm 0.01}$ | $\mathbf{0.22 \pm 0.01}$ | $0.70 \pm 0.02$ | $0.11 \pm 0.00$ | $0.61 \pm 0.01$ | $0.04 \pm 0.01$ |
| 0.50 | **1.00** | 71.52 | $\mathbf{0.32 \pm 0.01}$ | $0.23 \pm 0.01$ | $\mathbf{0.72 \pm 0.02}$ | $0.11 \pm 0.00$ | $0.63 \pm 0.02$ | $0.05 \pm 0.01$ |
| 0.40 | **1.00** | 66.58 | $0.31 \pm 0.01$ | $0.24 \pm 0.01$ | $0.66 \pm 0.03$ | $0.12 \pm 0.01$ | $0.66 \pm 0.01$ | $0.05 \pm 0.01$ |
| 0.30 | 0.99 | 52.09 | $0.31 \pm 0.01$ | $0.24 \pm 0.01$ | $0.65 \pm 0.03$ | $0.13 \pm 0.01$ | $0.65 \pm 0.02$ | $0.06 \pm 0.01$ |
| 0.20 | 0.97 | 21.97 | $0.29 \pm 0.01$ | $\mathbf{0.22 \pm 0.01}$ | $0.60 \pm 0.03$ | $0.11 \pm 0.01$ | $0.69 \pm 0.01$ | $0.08 \pm 0.01$ |
| 0.10 | 0.90 | 30.19 | $0.29 \pm 0.01$ | $0.24 \pm 0.01$ | $0.59 \pm 0.02$ | $0.10 \pm 0.00$ | $0.71 \pm 0.02$ | $0.08 \pm 0.02$ |
| 0.05 | 0.64 | 0.21 | $0.27 \pm 0.01$ | $\mathbf{0.22 \pm 0.01}$ | $0.54 \pm 0.01$ | $\mathbf{0.09 \pm 0.00}$ | $0.76 \pm 0.02$ | $\mathbf{0.11 \pm 0.01}$ |
| 0.01 | 0.61 | 0.14 | $0.26 \pm 0.00$ | $0.24 \pm 0.01$ | $0.53 \pm 0.01$ | $\mathbf{0.09 \pm 0.00}$ | $0.78 \pm 0.01$ | $0.07 \pm 0.00$ |

Table 21: Accuracy–manipulation trade-off for the GS attack on the "broccoli" output neuron in **ViT-L/32** . Reported are test accuracy (in %), AUROC for the "broccoli" class, and the mean ± standard deviation of alignment and similarity metrics, computed over 100 independent FV runs.

| | | | Alignment | | Target Similarity | | | |
|---|---|---|---|---|---|---|---|---|
| $\alpha$ | AUC | Acc. | Target Lbl. ↑ | GT Lbl. ↓ | CLIP ↑ | MSE ↓ | LPIPS ↓ | SSIM ↑ |
| Original | **1.00** | **76.97** | $0.23 \pm 0.02$ | $0.28 \pm 0.02$ | $0.51 \pm 0.02$ | $0.08 \pm 0.00$ | $0.73 \pm 0.03$ | $0.08 \pm 0.01$ |
| 0.999999 | **1.00** | 76.58 | $0.22 \pm 0.02$ | $0.28 \pm 0.02$ | $0.51 \pm 0.02$ | $0.08 \pm 0.00$ | $0.73 \pm 0.03$ | $0.08 \pm 0.01$ |
| 0.999900 | **1.00** | 76.64 | $\mathbf{0.29 \pm 0.03}$ | $0.27 \pm 0.03$ | $\mathbf{0.53 \pm 0.03}$ | $0.09 \pm 0.00$ | $0.74 \pm 0.04$ | $0.08 \pm 0.01$ |
| 0.999500 | 0.99 | 76.06 | $\mathbf{0.29 \pm 0.02}$ | $\mathbf{0.25 \pm 0.01}$ | $0.50 \pm 0.02$ | $0.07 \pm 0.00$ | $0.74 \pm 0.02$ | $0.08 \pm 0.01$ |
| 0.999000 | 0.99 | 75.96 | $0.28 \pm 0.02$ | $\mathbf{0.25 \pm 0.01}$ | $0.49 \pm 0.02$ | $0.07 \pm 0.00$ | $0.72 \pm 0.02$ | $0.09 \pm 0.01$ |
| 0.990000 | 0.99 | 72.42 | $\mathbf{0.29 \pm 0.02}$ | $\mathbf{0.25 \pm 0.01}$ | $0.49 \pm 0.02$ | $0.06 \pm 0.00$ | $\mathbf{0.71 \pm 0.01}$ | $0.11 \pm 0.01$ |
| 0.100000 | 0.50 | 0.10 | $0.25 \pm 0.01$ | $0.27 \pm 0.01$ | $0.49 \pm 0.01$ | $\mathbf{0.05 \pm 0.00}$ | $0.75 \pm 0.01$ | $\mathbf{0.18 \pm 0.01}$ |

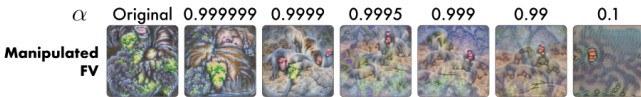

Figure 18: Sample FVs at different values of $\alpha$ for **ViT-L/32**.

## D.6 Impact of Target Image on Manipulation

We provide additional quantitative evaluation of the effect of the target image on manipulation success, extending the results from Sec. 4.4 to an additional set of target images, shown in Fig. 19.

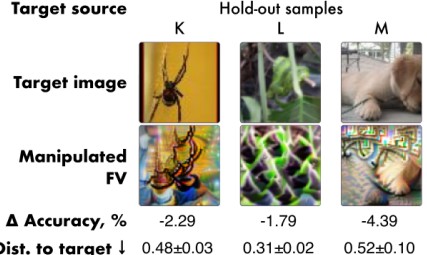

Figure 19: Manipulation results for different target images by source. Extended version of Table 13 including additional target images.

## D.7 Attack Detection

We provide additional qualitative results for our attack detection scheme in Fig. 20, extending the evaluation described in Sec. 6 to additional experimental settings.

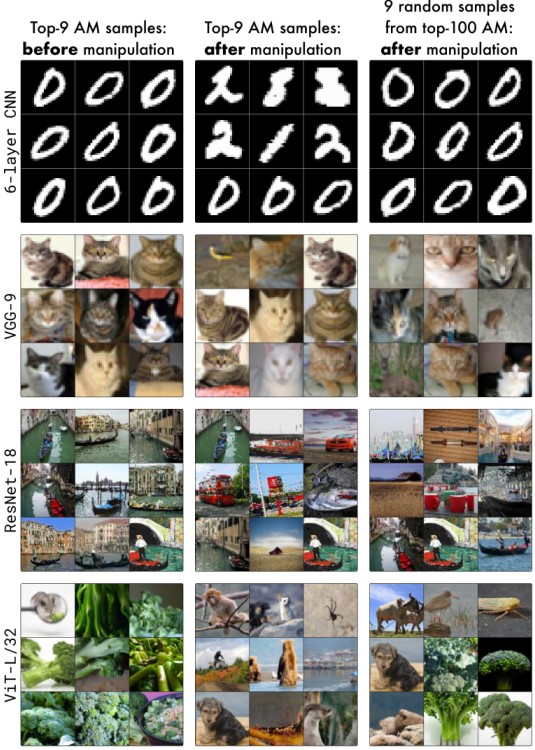

Figure 20: Top-9 most activating test samples before and after the GS attack. While the attack can substantially change which samples rank as the most activating across a dataset, broader sampling from the top of the AM distribution often recovers the original concept.

