# OpenReview forum: "Manipulating Feature Visualizations with Gradient Slingshots"
_NeurIPS.cc/2025/Conference — NeurIPS 2025 poster_

### Official Review · Reviewer_brRT · 2025-06-26

**Clarity:** 3
**Significance:** 3
**Originality:** 4
**Rating:** 5
**Confidence:** 4

**Summary:**

This paper proposes an attack on Activation Maximization methods, which use a parametrized image space to find images that maximize the dot product between the model's representation of the parameterized image and an arbitrary vector in the representation space. The theoretic basis of the method is that AM on the Euclidean distance between a parameter to be manipulated and a target parameter for AM to find will result in AM converging to the target parameter. In practice, the Gradient Slingshot optimizes a weighted combination of two loss terms. The preservation loss minimizes the change in model features and dot product with the arbitrary vector. The manipulation loss is ideally a second order optimization that changes the model's gradient to match the gradient of the Euclidean distance to the target parameter. For larger models, the function can be changed to match the Euclidean distance instead. Experiments show that AM can be manipulated with minimal changes to other model behavior. There is a trade-off between preserving behavior and strength of the AM manipulation. The attack is more effective on models with more parameters. AM on test set images are also mostly consistent before and after manipulation, making the attack difficult to detect.

**Questions:**

Q1: I think Eq. 4 needs to be updated to use the negative square Euclidean distance instead of the positive distance. Currently the gradient with respect to Eq. 4 would result in $-1$ times Eq. 3.

Q2: How were the standard deviation values calculated? Multiple trained models? Bootstrapping?

Q3: As detailed in W3, how was hyperparameter tuning performed?

**Ethical Concerns:**

["NO or VERY MINOR ethics concerns only"]

**Final Justification:**

The authors have sufficiently addressed my concerns and I appreciate the additional details provided. My perspective has not changed with respect to the significance of Lemma 3.1, but I do not think this holds the paper back. I continue to recommend the paper to be accepted.

**Limitations:**

Yes.

**Quality:**

4

**Strengths And Weaknesses:**

S1: The Gradient Slingshot is a novel attack on AM that, unlike previous methods, does not change the model architecture. This makes the attack harder to detect, as confirmed in Section 4.3.

S2: The sampling procedure for the manipulation loss is a unique way to manipulate behavior for a subset of features.

S3: The experiments are well documented, show good results, and include the standard deviation.

W1: The paper seems related to other attacks, such as the proposed method sort of forcing AM to be adversarial image perturbation. The related works section does not discuss this broader context and I'm curious how the proposed method relates.

W2: Lemma 3.1, used as the theoretical basis for the method, is a very mundane result that gradient descent (see Q1) on Euclidean distance will converge. The most novel part of the result is establishing that $\mathbb{M} = \mathbb{T}_{B, L}$, but the proof uses that as a condition instead of emphasizing that fact.

W3: Few details are given for the hyperparameter tuning process besides that it was computationally intensive. Was it done via grid search on a validation set? How many models needed to be trained during tuning?

---

> ### Author Rebuttal · Authors · 2025-07-31
>
> We thank the reviewer for their thorough and constructive feedback. We are glad they found the Gradient Slingshots (GS) attack to be novel and difficult to detect, and appreciated the thoroughness and convincing nature of our experimental results. Below, we address the reviewer’s concerns point by point.
>
> ### Weaknesses
>
> > W2: Lemma 3.1, used as the theoretical basis for the method, is a very mundane result that gradient descent (see Q1) on Euclidean distance will converge. The most novel part of the result is establishing that $\mathbb{M} = \mathbb{T_{\text{B, L}}}$, but the proof uses that as a condition instead of emphasizing that fact.
>
> We clarify that since we control how the manipulation set $\mathbb{M}$ is sampled, we explicitly define it as $\mathbb{T_{\text{B, L}}}$; the non-trivial aspect lies in ensuring that each step $q^{(i)}$ during optimization remains in $\mathbb{T_{\text{B, L}}}$, which is guaranteed for gradient descent but not necessarily for other optimizers. This point is already noted in lines 149–151, where we explain that while Lemma 3.1 formally applies to gradient descent, our empirical results indicate that similar behavior extends to other Feature Visualization (FV) optimization strategies.
>
> > W3: Few details are given for the hyperparameter tuning process besides that it was computationally intensive. Was it done via grid search on a validation set? How many models needed to be trained during tuning?
>
> We thank the reviewer for raising this concern. A detailed description of our hyperparameter tuning and selection procedure is provided in our response to Question 3 below, and we have incorporated this explanation into the revised paper.
>
> ### Questions
>
> > Q1: I think Eq. 4 needs to be updated to use the negative square Euclidean distance instead of the positive distance. Currently the gradient with respect to Eq. 4 would result in times Eq. 3.
>
> We thank the reviewer for pointing out this oversight. We have corrected the Eq. 4 accordingly. We have also updated Equation 10 to fix the same issue. We emphasize that this correction only affects the presentation in the manuscript; the code implementation already uses the correct formulation and remains unaffected.
>
> > Q2: How were the standard deviation values calculated? Multiple trained models? Bootstrapping?
>
> As detailed in Appendix C.8, the reported standard deviations are computed over multiple runs of FV generation. Specifically, we generated 100 samples per model (30 for CLIP ViT-L/14 due to computational constraints), and computed similarity metrics over these samples. Since FV generation involves sampling the initialization point from a distribution, each run yields a slightly different result. The standard deviations thus characterize the consistency of the metrics across these independently generated FVs. We have incorporated this clarification into all relevant table captions to prevent any confusion.
>
> > Q3: As detailed in W3, how was hyperparameter tuning performed?
>
> For the parameter $\alpha$, our tuning process follows a coarse-to-fine iterative approach. We begin with a broad sweep of values (e.g., $\alpha \in [0.01, 0.05, 0.5, 0.9]$) to identify a promising region. Once a candidate such as $\alpha = 0.5$ shows good results, we perform a finer-grained search around it (e.g., 0.4 and 0.6), and continue refining around the best-performing values, e.g., near 0.6.
>
> Importantly, we perform tuning using only 1–3 epochs and on small subsets of large datasets (see Table 8 in the Appendix). For ViT experiments, for example, we use less than 1\% of ImageNet, with a single epoch taking under 5 minutes. While we acknowledge that tuning of $\alpha$ is computationally intense, the process remains practical even with modest resources.
>
> For the parameter $w$, we set it to 0.1 when the target layer contains 10 neurons, and to 0.01 when it contains 1000 neurons, preventing the target neuron's activation landscape from being overly disrupted. For $\gamma$, we use the heuristic $\nabla (\phi \circ \eta)(\mathbf{0}) = \gamma \mathbf{q^t}$ to preserve the gradient magnitude in the initialization zone.
>
> We have incorporated this explanation into the paper in order to improve transparency around hyperparameter selection.

---

> ### Comment · Area_Chair_sC5n · 2025-08-05
>
> Please let the authors know whether their rebuttal has adequately addressed your concerns. If any issues remain, please communicate your specific, unresolved concerns as soon as possible to ensure timely discussion.

---

### Official Review · Reviewer_zpF6 · 2025-06-27

**Clarity:** 4
**Significance:** 3
**Originality:** 3
**Rating:** 5
**Confidence:** 4

**Summary:**

This paper addresses attacks on feature visualization. In a scenario where model developers want to conceal the true feature visualization of certain features from a third-party auditor, their method can modify any feature’s feature visualization to any target image by adjusting the model parameters. This ensures that the model’s performance remains intact, while the feature visualization of the target feature resembles the input image. Essentially, they can alter any single feature visualization to anything they desire.

Their method fine-tunes the model using a loss compromise consisting of two parts. The first part aims to maintain the activations on unrelated elements unchanged. The second part directs the gradients of data points in a tunnel between the feature visualization initialization points and the target input image towards the target image. They conducted numerous empirical studies to assess the impact of model size, target image familiarity to the model, and other factors.

**Questions:**

## Questions
 Is there an experiment that shows the effect of model manipulation on other features feature visualization ? Do other features FV also look like the target image?

**Ethical Concerns:**

["NO or VERY MINOR ethics concerns only"]

**Final Justification:**

1) They showed that they really don't change the model behaviour in their rebuttal.
2) They make the hyperparameter tuning process more clear.

**Limitations:**

Yes

**Paper Formatting Concerns:**

No issue.

**Quality:**

3

**Strengths And Weaknesses:**

## Weakness
1. The paper only focuses on last layer features. It does not conduct any experiments on features in middle layers.
2. Hyperparameter tuning of loss seems very hard as both large and small values lead to the method not working.
3. The paper claims that internal mechanisms of the model remain unchanged but because they didn’t investigate middle layers their argument is only based on accuracy.

## Strengths
1. Good writing.
2. Novel idea for Feature visualization manipulation.
3. Good variation of model choices.
4. Many ablation studies.

---

> ### Author Rebuttal · Authors · 2025-07-31
>
> We thank the reviewer for their clear summary and thoughtful feedback. We are glad they found our paper well-written, our idea novel, and appreciated the breadth of model choices and ablations. Below, we address the reviewer’s concerns in detail.
>
> ### Weaknesses
>
> > 1. The paper only focuses on last layer features. It does not conduct any experiments on features in middle layers.
>
> While Section 4 primarily focuses on output-layer features to enable direct quantitative evaluation (e.g., via AUROC against known ground-truth labels), our Gradient Slingshots (GS) method is not limited to the final layer. The general formulation in Section 3 applies to any differentiable feature function within the network.
>
> Moreover, the submitted version of our paper already includes a case study involving the manipulation of a mid-layer feature (Section 5). Specifically, we target a neuron in the 22nd residual block of a CLIP ViT-L/14-based model, identified through a supervised discovery method. Both quantitative results (Table 3) and qualitative visualization (Figure 1) demonstrate that GS manipulation is effective when applied to middle layers, and preserves the feature performance.
>
> > 2. Hyperparameter tuning of loss seems very hard as both large and small values lead to the method not working.
>
> We thank the reviewer for raising this concern. For the parameter $\alpha$, our tuning process follows a coarse-to-fine iterative approach. We begin with a broad sweep of values (e.g., $\alpha \in [0.01, 0.05, 0.5, 0.9]$) to identify a promising region. Once a candidate such as $\alpha = 0.5$ shows good results, we perform a finer-grained search around it (e.g., 0.4 and 0.6), and continue refining around the best-performing values, e.g., near 0.6.
>
> Importantly, we perform tuning using only 1–3 epochs and on small subsets of large training datasets (see Table 8 in the Appendix). For ViT experiments, for example, we use less than 1\% of ImageNet, with a single epoch taking under 5 minutes. While we acknowledge that tuning of $\alpha$ is computationally intense, the process remains practical even with modest resources.
>
> For the parameter $w$, we set it to 0.1 when the target layer contains 10 neurons, and to 0.01 when it contains 1000 neurons – preventing the target neuron's activation landscape from being overly disrupted. For $\gamma$, we use the heuristic $\nabla (\phi \circ \eta)(\mathbf{0}) = \gamma \mathbf{q^t}$ to preserve the gradient magnitude in the initialization zone.
>
> We have incorporated this explanation into the paper in order to improve transparency around hyperparameter selection.
>
>
> > 3. The paper claims that internal mechanisms of the model remain unchanged but because they didn’t investigate middle layers their argument is only based on accuracy.
>
> As noted earlier, we explicitly evaluate manipulation on a feature located in an intermediate layer (Section 5, Case Study with CLIP ViT-L/14). In all our quantitative experiments, beyond overall accuracy, we measure the AUROC of the manipulated feature output with respect to its ground truth label. This targeted metric directly assesses whether the feature’s original performance is preserved. Our key argument is that the target feature is the component most susceptible to alteration by our manipulation. The fact that it maintains its original predictive behavior demonstrates that the model’s reliance on this feature remains intact, and that our method effectively “covers up” the FV without removing or degrading its internal role. We have made this argument more prominent and explicit in the paper.
>
> ### Questions
> > Is there an experiment that shows the effect of model manipulation on other features Feature Visualization ? Do other features FV also look like the target image?
>
> We understand the reviewer’s concern — it is important to demonstrate that GS does not inadvertently perturb FVs of non-manipulated features, which could make GS detectable. To address this, we evaluated FVs of the first 10 non-manipulated output neurons in a GS-attacked ResNet-50 (Sec. 4, $\alpha=0.64$). For each neuron, we sampled 100 FVs from the original and manipulated models and measured their similarity to a single original FV and the target image.
>
> The results, presented in Table C below, show virtually no difference in the FVs of non-manipulated neurons between the two models across all metrics, confirming that GS leaves non-manipulated features unaffected. We have included these quantitative results, along with example FVs, to the paper’s Appendix.
>
> **Table C: Before-and-after-manipulation comparison of visualizations of non-manipulated features in the GS attack on the ResNet-50 model where the “wolf spider” output neuron was manipulated. Reported are the mean ± standard deviation of similarity between FVs and a single sampled original FV (CLIP, MSE, LPIPS), and the mean ± standard deviation of similarity between manipulated FV and the target image. Similarity statistics are computed over 100 independent FV runs. These results demonstrate that the GS attack does not perturb the visualization of non-manipulated features.**
>
> | Neuron | Model         | CLIP $\uparrow$   | MSE $\downarrow$   | SSIM $\uparrow$      | CLIP $\uparrow$   | MSE $\downarrow$   | SSIM $\uparrow$   |
> |----------------:|:--------------|:----------------|:---------------|:----------------|:------------------|:-------------------|:------------------|
> | | | Original FV     |       |        | Target  |  |
> |               0 | Original    | $0.89\pm0.03$   | $0.18\pm0.02$  | $0.03\pm0.01$   | $0.54\pm0.02$     | $0.14\pm0.01$      | $0.05\pm0.01$     |
> |                | Manipulated | $0.90\pm0.03$   | $0.18\pm0.01$  | $0.03\pm0.01$   | $0.55\pm0.02$     | $0.14\pm0.01$      | $0.06\pm0.01$     |
> |               1 | Original    | $0.94\pm0.02$   | $0.16\pm0.01$  | $0.03\pm0.01$   | $0.53\pm0.02$     | $0.13\pm0.01$      | $0.05\pm0.01$     |
> |               | Manipulated | $0.94\pm0.02$   | $0.17\pm0.01$  | $0.03\pm0.01$   | $0.54\pm0.02$     | $0.13\pm0.01$      | $0.06\pm0.01$     |
> |               2 | Original    | $0.95\pm0.02$   | $0.18\pm0.01$  | $0.03\pm0.01$   | $0.53\pm0.02$     | $0.13\pm0.01$      | $0.05\pm0.01$     |
> |               | Manipulated | $0.96\pm0.01$   | $0.18\pm0.01$  | $0.03\pm0.01$   | $0.53\pm0.01$     | $0.14\pm0.01$      | $0.06\pm0.01$     |
> |               3 | Original    | $0.93\pm0.03$   | $0.19\pm0.02$  | $0.03\pm0.01$   | $0.54\pm0.01$     | $0.15\pm0.01$      | $0.06\pm0.01$     |
> |               | Manipulated | $0.90\pm0.02$   | $0.19\pm0.01$  | $0.03\pm0.01$   | $0.54\pm0.01$     | $0.14\pm0.01$      | $0.06\pm0.01$     |
> |               4 | Original    | $0.95\pm0.01$   | $0.17\pm0.01$  | $0.03\pm0.01$   | $0.52\pm0.02$     | $0.14\pm0.01$      | $0.05\pm0.01$     |
> |               | Manipulated | $0.82\pm0.09$   | $0.16\pm0.01$  | $0.03\pm0.01$   | $0.53\pm0.01$     | $0.14\pm0.01$      | $0.06\pm0.01$     |
> |               5 | Original    | $0.95\pm0.02$   | $0.15\pm0.01$  | $0.03\pm0.01$   | $0.52\pm0.02$     | $0.13\pm0.01$      | $0.05\pm0.01$     |
> |               | Manipulated | $0.93\pm0.02$   | $0.14\pm0.01$  | $0.03\pm0.01$   | $0.52\pm0.02$     | $0.12\pm0.01$      | $0.05\pm0.01$     |
> |               6 | Original    | $0.96\pm0.02$   | $0.15\pm0.01$  | $0.03\pm0.01$   | $0.53\pm0.02$     | $0.12\pm0.01$      | $0.05\pm0.01$     |
> |               | Manipulated | $0.96\pm0.01$   | $0.15\pm0.01$  | $0.03\pm0.01$   | $0.52\pm0.01$     | $0.12\pm0.01$      | $0.05\pm0.01$     |
> |               7 | Original    | $0.92\pm0.03$   | $0.18\pm0.01$  | $0.03\pm0.01$   | $0.55\pm0.02$     | $0.13\pm0.01$      | $0.05\pm0.01$     |
> |               | Manipulated | $0.94\pm0.02$   | $0.19\pm0.01$  | $0.03\pm0.01$   | $0.56\pm0.02$     | $0.13\pm0.01$      | $0.05\pm0.01$     |
> |               8 | Original    | $0.96\pm0.01$   | $0.19\pm0.01$  | $0.03\pm0.01$   | $0.57\pm0.02$     | $0.13\pm0.01$      | $0.05\pm0.01$     |
> |               | Manipulated | $0.96\pm0.01$   | $0.20\pm0.01$  | $0.03\pm0.01$   | $0.58\pm0.01$     | $0.13\pm0.01$      | $0.05\pm0.01$     |
> |               9 | Original    | $0.93\pm0.02$   | $0.17\pm0.01$  | $0.03\pm0.01$   | $0.49\pm0.02$     | $0.13\pm0.01$      | $0.05\pm0.01$     |
> |               | Manipulated | $0.92\pm0.02$   | $0.17\pm0.01$  | $0.02\pm0.01$   | $0.51\pm0.02$     | $0.13\pm0.01$      | $0.05\pm0.01$     |

---

> > ### Comment · Reviewer_zpF6 · 2025-08-05
> >
> > Thanks for your reponse. You addressed my all of my concerns. I will update my score accordingly.

---

> ### Comment · Area_Chair_sC5n · 2025-08-05
>
> Please let the authors know whether their rebuttal has adequately addressed your concerns. If any issues remain, please communicate your specific, unresolved concerns as soon as possible to ensure timely discussion.

---

### Official Review · Reviewer_PQ6K · 2025-07-01

**Clarity:** 3
**Significance:** 3
**Originality:** 3
**Rating:** 4
**Confidence:** 4

**Summary:**

The paper introduces Gradient Slingshots (GS), a method that modifies a model so that gradient ascent, involved in FV, is drawn towards a wrong optimum. The optimum, which is an image, can be arbitrarily chosen beforehand by the neural network designer, who would seek to fool an audit based on feature visualization. As a result, the result of the FV looks like the wrong optimum rather than the actual FV, potentially hiding the actual concepts underlying the neural network's predictions. GS applies as a fine-tuning that is supposed to preserve the model's predictive capacities and alter only the behaviour of FV's gradient ascent.

The authors test their method on simple CN, VGG, ResNet18/50, and ViT, on some neurons involved in Cifar10 and Imagenet classification. They also provide a case study based on CLIP and weapon classification to emphasize the risks of such an attack.

**Questions:**

- How can a manipulator make sure that the FV initialization will fall into $\mathbb{B}$ and will be appropriately steered towards the wrong optimum?
- How does GS affect the FV of other features? If the FV is pertubated for every neurons, it might seem suspicious to auditers.
- In Sec 5, the manipulation and the FV seem to target the feature vector representing an assault rifle. However, FV and GS are only formulated for the maximization of a single neuron. How to make this fall under a formalism similar to Eq. 1, 2, 3, 4, 5? If the answer is contained in lines 290 -> 293, it is not detailed enough, and more description should be provided for reproducibility.
- The FV method from Olah et al. 2016 [1] notoriously fails on modern networks like ViT [2]. Could the authors run GS with modern FV methods like MACO [2]
- I am surprised that there is no factor $2$ in Eq. 10 before $\gamma$, or a factor $\frac{1}{2}$ in Eq. 9, that stems from derivatives of square expressions.

[1] Distill, Olah et al., 2017 [https://distill.pub/2017/feature-visualization/](https://distill.pub/2017/feature-visualization/)
[2] Unlocking Feature Visualization for Deeper Networks with MAgnitude Constrained Optimization, Fel et al. NeurIPS 2023

**Ethical Concerns:**

["NO or VERY MINOR ethics concerns only"]

**Final Justification:**

After careful thinking and considering authors-reviewers discussions:

-   I am worried about the current impact of the work since FV ibased auditing is not (yet) a common practice. But I recognize that AI regulation is an increasing trend and is likely to become even more important in the future. I think that it is a sound direction for AI, and that NeurIPS should support it.
-   The generalization of GS to any image, for which we only have access to quantitative evaluation, is a concern, but if the authors include more qualitative examples in a revised version and these visualizations are compelling, I deem that the article will be worth publishing at NeurIPS.

Therefore, I increased my rating to 4 and strongly encouraged the authors to add a substantial visualization gallery to their work.

**Limitations:**

yes

**Paper Formatting Concerns:**

No formatting problems detected.

**Quality:**

3

**Strengths And Weaknesses:**

### Strengths

- The idea is novel and simple
- The paper is well written - even if it could be slightly rebalanced for improved readability, see below)
- The manipulation is undetectable by looking at the code or the structure of the model since it is contained in the weights' values.

### Weaknesses

- **W1**: The evaluation is only conducted on one or a few neurons, so it might be subject to cherry-picking.
- **W2**: The quantitative metrics are not impressive. I am suspicious that GS works on only a few examples selected for the paper illustrations, but statistically, GS seems to only slightly improve the similarity metrics compared to the original FV. I am not convinced that it actually works. It is even more true for ViT (see Appendix, p. 32), for which the metrics barely change compared to the FV without GS.
- **W3**: The assumption of l. 139 for Lemma 3.1 seems to make GS artificial. As far as I understand, it means that the initialization must be known by the manipulator and fall in $\mathbb{B}$. It seems very easy to circumvent for an auditor, for instance, by uniformly sampling several initializations in $\mathcal{Q}$.

### Comments and suggestions

- The assumption of l. 139 for Lemma 3.1 seems to be very strong, but it is only stated in the proof. It should appear in the Lemma.
- The authors spend time on the proof of Lemma 3.1, which is intuitive, so they do not need proof to make sense of it. Hence, it takes up some unnecessary space, which could be better used for explaining the metrics or expanding the case study setting if it was deferred to the appendix.

---

> ### Author Rebuttal · Authors · 2025-07-31
>
> We thank the reviewer for their thoughtful feedback. We’re pleased they found our method novel and simple, our paper well-written, and appreciated that our attack is difficult to detect. Below, we address their concerns to clarify that our results are comprehensive and not selectively presented, provide additional quantitative evidence, discuss theoretical assumptions, and elaborate on broader applicability.
>
> ### Weaknesses
> > W1: The evaluation is only conducted on one or a few neurons, so it might be subject to cherry-picking.
>
> We thank the reviewer for raising concerns about cherry-picking. To address this, we further manipulated the first 10 output neurons in the ResNet-50 setting from Sec. 4 (target image: Dalmatian). Beyond standard similarity metrics, we added two semantic alignment metrics:
>
> - Target Label Alignment ("Target Lbl."): CLIP cosine similarity between manipulated FV and the target label embeddings (e.g., “an image of a Dalmatian”).
> - GT Label Alignment ("GT Lbl."): CLIP similarity between manipulated FV and the ground-truth feature label (e.g., “an image of a wolf spider”).
>
> Results from Table A (only the first 5 neurons shown due to character limits) show that manipulated FVs are similar to the target image and align more with the target than ground truth label; classification accuracy and AUROC indicate the stability of model and feature performance. This supports that our method generalizes beyond isolated neurons.
>
> **Table A: Manipulation results for the Gradient Slingshots (GS) attack on the first 10 output neurons of a ResNet-50 (partial, 5/10 neurons).**
>
> | Neuron | Model  | Acc.       | AUROC | Target Lbl.↑      | GT Lbl.↓        | CLIP ↑          |
> |--------|--------|------------|-------|-------------------|-----------------|-----------------|
> | 0      | Orig.  | **76.13**  | 1     | 0.22 ± 0.01       | 0.31 ± 0.01     | 0.55 ± 0.02      |
> |        | Manip. | 75.23      | 1     | **0.32 ± 0.01**   | **0.21 ± 0.01** | **0.67 ± 0.02**  |
> | 1      | Orig.  | **76.13**  | 1     | 0.23 ± 0.01       | 0.28 ± 0.01     | 0.53 ± 0.02      |
> |        | Manip. | 75.15      | 1     | **0.32 ± 0.01**   | **0.25 ± 0.01** | **0.68 ± 0.02**  |
> | 2      | Orig.  | **76.13**  | 1     | 0.24 ± 0.01       | 0.28 ± 0.02     | 0.53 ± 0.02      |
> |        | Manip. | 75.17      | 1     | **0.33 ± 0.01**   | **0.23 ± 0.01** | **0.72 ± 0.02**  |
> | 3      | Orig.  | **76.13**  | 1     | 0.24 ± 0.01       | 0.27 ± 0.02     | 0.54 ± 0.01      |
> |        | Manip. | 75.18      | 1     | **0.32 ± 0.01**   | **0.22 ± 0.01** | **0.69 ± 0.02**  |
> | 4      | Orig.  | **76.13**  | 1     | 0.23 ± 0.01       | 0.28 ± 0.01     | 0.52 ± 0.01      |
>
> > W2: The quantitative metrics are not impressive. I am suspicious that GS works on only a few examples selected for the paper illustrations, [...] It is even more true for ViT (see Appendix, p. 32), for which the metrics barely change compared to the FV without GS.
>
> We thank the reviewer for raising the concern. As noted in Sec. 4 and Appendix B, low-level metrics (MSE, SSIM, LPIPS) are more useful in settings with low-res datasets (e.g., MNIST, CIFAR), where manipulated FVs resemble targets pixel-wise. In high-res settings, these metrics fail to capture compositional or semantic alignment (see Fig. 3). CLIP similarity better reflects high-level alignment but may still miss compositional effects.
>
> To address the selective representation concerns, we added multiple samples of manipulated FVs per evaluation. We also extended all quantitative evaluations using Target/GT Label Alignment metrics, capturing semantic similarity. Table B (extending Table 16) for ViT-L/32 shows that at some $\alpha$ values, FVs align more with the target image label (“sea lions”) than the ground truth (“broccoli”), supporting GS’s manipulation efficacy on transformer models.
>
> **Table B: Accuracy–manipulation trade-off for the GS attack on the ``broccoli'' output neuron in ViT-L/32 (partial Tab. 16).**
>
> | α        | Acc.       | Target Lbl.↑       | GT Lbl.↓          |
> |----------|------------|--------------------|-------------------|
> | Orig.    | **76.97**  | 0.22 ± 0.03        | 0.28 ± 0.02       |
> | 0.999999 | 76.58      | 0.23 ± 0.02        | 0.28 ± 0.02       |
> | 0.9999   | 76.64      | **0.29 ± 0.02**    | 0.27 ± 0.03       |
> | 0.9995   | 76.06      | **0.29 ± 0.02**    | **0.25 ± 0.01**   |
> | 0.999    | 75.96      | 0.28 ± 0.02        | **0.25 ± 0.01**   |
> | 0.99     | 72.42      | **0.29 ± 0.02**    | **0.25 ± 0.01**   |
> | 0.1      | 0.1        | 0.25 ± 0.01        | 0.26 ± 0.01       |
>
> > W3: The assumption of l. 139 for Lemma 3.1 seems to make GS artificial. As far as I understand, it means that the initialization must be known by the manipulator and fall in. It seems very easy to circumvent for an auditor, for instance, by uniformly sampling several initializations in $Q$.
>
> We thank the reviewer and agree the assumption in l.139 may appear strong. We provide further explanation for why this assumption is reasonable in our response to Q1.
>
> >The assumption of l. 139 for Lemma 3.1 […] should appear in the Lemma.
> >The authors spend time on the proof of Lemma 3.1, which is intuitive, […].
>
> We thank the reviewer for the suggestion. The assumption is now stated explicitly in Lemma 3.1, and the detailed proof has been moved to the Appendix. The freed space was used to clarify evaluation metrics and hyperparameter selection.
>
> ### Questions
>
> > 1. How can a manipulator make sure that the FV initialization will fall into and will be appropriately steered towards the wrong optimum?
>
> We thank the reviewer for the question. Standard FV libraries (Lucid, Lucent, Horama) use fixed Gaussian noise initialization, making the process predictable. While auditors could randomize or broaden initialization, adversaries can expand the “slingshot zone” $\mathbb{B}$ accordingly. Alternative methods (e.g., seeding from real images) are rare since they often yield FVs resembling the seed (Hamblin et al., 2024).
>
> - Hamblin et al. “Feature Accentuation: Revealing 'What' Features Respond to in Natural Images.”
>
> > 2. How does GS affect the FV of other features? If the FV is pertubated for every neurons, it might seem suspicious to auditors.
>
> We appreciate the reviewer’s concern and agree it’s important to show GS doesn’t alter FVs of non-manipulated features. To address this, we evaluated FVs of the first 10 non-manipulated output neurons in a GS-attacked ResNet-50 (Sec. 4, $\alpha=0.64$). For each neuron, we sampled 100 FVs from the original and manipulated models and measured their similarity to a single original FV and the target image.
>
> Table C (first 5 neurons shown due to length limits) shows that non-manipulated neuron FVs remain nearly unchanged, confirming that GS preserves non-manipulated features. Qualitative and quantitative results are now included in the Appendix.
>
> **Table C: Before-and-after-manipulation comparison of visualizations of non-manipulated features in the GS attack on the ResNet-50 model (partial, 5/10 neurons).**
>
> | Neuron | Model  | CLIP Orig. FV  | CLIP Target Img.|
> |--------|--------|--------------|-------------|
> | 0     | Orig.  | 0.89 ± 0.03 | 0.54 ± 0.02     |
> |        | Manip. | 0.90 ± 0.03  | 0.55 ± 0.02     |
> | 1     | Orig.  | 0.94 ± 0.02 | 0.53 ± 0.02     |
> |        | Manip. | 0.94 ± 0.02  | 0.54 ± 0.02     |
> | 2     | Orig.  | 0.95 ± 0.02  | 0.53 ± 0.02     |
> |        | Manip. | 0.96 ± 0.01 | 0.53 ± 0.01     |
> | 3     | Orig.  | 0.93 ± 0.03  | 0.54 ± 0.01     |
> |        | Manip. | 0.90 ± 0.02  | 0.54 ± 0.01     |
> | 4     | Orig.  | 0.95 ± 0.01  | 0.52 ± 0.02     |
> |        | Manip. | 0.82 ± 0.09   | 0.53 ± 0.01     |
>
> > 3. In Sec 5, the manipulation and the FV seem to target the feature vector representing an assault rifle. However, FV and GS are only formulated for the maximization of a single neuron. How to make this fall under a formalism similar to Eq. 1, 2, 3, 4, 5?
>
> We thank the reviewer for highlighting this point. Both the theory and implementation (Eqs. 1–5) define GS manipulation over a feature function $f$, not just individual neurons. As stated in l. 68–70, $f(\mathbf{x}):= \mathbf{v} \cdot g(\mathbf{x})$, where $g(\mathbf{x})$ is a vector of activations and $\mathbf{v}$ defines a feature direction.
>
> > 4. The FV method from Olah et al. 2016 [1] notoriously fails on modern networks like ViT [2]. Could the authors run GS with modern FV methods like MACO [2]
>
> We thank the reviewer for the suggestion. While we show that standard Fourier FV with an appropriate transformation robustness recipe (Appendix C.7) can produce interpretable results for modern architectures like CLIP ViT-L/14 (Fig. 1), we agree that MACO is a popular FV approach for transformers.
>
> MACO, like Fourier FV (Olah et al., 2017), operates in the Fourier frequency domain, but with a restricted optimization space, making it compatible with GS. We applied MACO to the existing manipulated ViT-L/32 model (Sec. 4), generating FVs before and after manipulation and evaluating alignment to the target and ground truth labels. As shown in Table D, manipulated MACO FVs show stronger alignment with the target class (“sea lions”) than the ground-truth (“broccoli”), mirroring results with Fourier FVs. We have included both quantitative and qualitative MACO results in the revised manuscript.
>
> **Table D: Manipulation results of GS applied to MACO visualizations of the “broccoli” output neuron in ViT-L/32.**
>
> |       | GT Lbl. ↓   | Target Lbl. ↑  |
> |-------|-----------|--------------|
> | Orig.   | 0.322 ± 0.029  | 0.264 ± 0.031 |
> | Manip.  | **0.242 ± 0.010** | **0.325 ± 0.012** |
>
> > 5. I am surprised that there is no factor in Eq. 10 before , or a factor in Eq. 9, that stems from derivatives of square expressions.
>
> We thank the reviewer for noticing this oversight on our side. We have corrected the Eq. 4 and the Eq. 10 accordingly. This error is limited only to the paper formalisms and not reflected in our code implementation.

---

> > ### Comment · Reviewer_PQ6K · 2025-08-04
> >
> > We would like to thank the authors for their thorough response, I appreciate the efforts put into the rebuttal.
> >
> > Some of my concerns were appropriately addressed, but I still find that:
> > - The impact of the work seems limited, since FV-based model auditing is not a common practice.
> > - The results are not very convincing, as it is difficult to assess the human plausibility of visualizations using quantitative metrics. Would an auditor be fooled by a manipulated FV that has a CLIP alignment of 0.32 with the desired image? The visualizations of the paper help to answer, but there are only a few, so it is difficult to assess the consistency of the method.

---

> > > ### Author Response · Authors · 2025-08-04
> > >
> > > Thank you for your thoughtful comments.
> > >
> > > > The impact of the work seems limited, since FV-based model auditing is not a common practice.
> > >
> > > While FV-based auditing is not yet widespread, this is largely because model auditing itself remains an emerging practice. Feature Visualization is one of the most established tools in interpretability [1–3], and one of the core motivations of interpretability research is to support trustworthy and rigorous model audits. It has also been argued that interpretability-based audits can offer stronger assurances about a model’s knowledge and capabilities than alternative, black-box approaches [4]. This makes our finding that FV-based explanations can be manipulated particularly impactful.
> > >
> > > [1] Samek, Wojciech, et al., eds. Explainable AI: Interpreting, explaining and visualizing Deep Learning. Vol. 11700. Springer Nature, 2019.
> > >
> > > [2] Bereska, Leonard, and Stratis Gavves. Mechanistic Interpretability for AI Safety – A Review. Transactions on Machine Learning Research, 2024.
> > >
> > > [3] Kazmierczak, Rémi, et al. "Explainability and vision foundation models: A survey." Information Fusion 122 (2025): 103184.
> > >
> > > [4] Casper, Stephen, et al. "Black-box access is insufficient for rigorous AI audits." Proceedings of the 2024 ACM Conference on Fairness, Accountability, and Transparency. 2024.
> > >
> > > > Would an auditor be fooled by a manipulated FV that has a CLIP alignment of 0.32 with the desired image?
> > >
> > > While a CLIP alignment score of 0.32 with the target concept (Table A in this rebuttal) may not appear high, we note that this value is typical for CLIP similarity between an image and its rough text description. For example, the alignment between the target image in this setting (a Dalmatian in Fig. 3) and its corresponding concept ("an image of a Dalmatian") is only 0.33. Moreover, the CLIP alignment score of a manipulated FV with the target (0.32) is significantly higher than the alignment with the feature’s actual concept in the same setting (e.g., 0.21 for neuron 0, Table A), indicating that the "fooling" is successful from CLIP's perspective. We will add this explanation to our manuscript.
> > >
> > > > The visualizations of the paper help to answer, but there are only a few, so it is difficult to assess the consistency of the method.
> > >
> > > We will include substantially more qualitative examples in the Appendix to better illustrate the consistency of our method.

---

> > > > ### Comment · Reviewer_PQ6K · 2025-08-05
> > > >
> > > > Thank you for your answer. I would like to remind that I find the paper interesting, well written, and elegant.
> > > >
> > > > After careful thinking:
> > > > - I am worried about the current impact of the work since FV ibased auditing is not (yet) a common practice. But I recognize that AI regulation is an increasing trend and is likely to become even more important in the future. I think that it is a sound direction for AI, and that NeurIPS should support it.
> > > > - The generalization of GS to any image, for which we only have access to quantitative evaluation, is a concern, but if you include more qualitative examples in a revised version and these visualizations are compelling, I deem that the article will be worth publishing at NeurIPS.
> > > >
> > > > Therefore, I increase my rating to 4 and strongly encourage the authors to add a substantial visualization gallery to their work.

---

### Official Review · Reviewer_Tk8Z · 2025-07-04

**Clarity:** 1
**Significance:** 1
**Originality:** 3
**Rating:** 2
**Confidence:** 4

**Summary:**

This paper introduces "Gradient Slingshots" (GS), a proposed attack method targeting Feature Visualization (FV), an explainable AI technique. The core premise is that if FV is used to audit AI models for fairness or other properties, an adversary could subtly manipulate their model's training to conceal malicious intent. The authors claim GS improves on prior methods by being architecture-agnostic (including transformers) and by maintaining the manipulated model's performance. The technique reportedly achieves this balance by optimizing a trade-off between the FV image's resemblance to a target and the model's original behavior, controlled by parameters alpha and w.

**Questions:**

Questions:

1. Could the authors clarify the meaning of the standard deviations reported throughout the paper? Are they based on 95% confidence intervals from multiple experimental runs? If so, what is the number of runs? Alternatively, are they computed for a single run or a single FV?

2. Is there a methodology for determining reasonable hyperparameter values without resorting to an exhaustive grid search? What would the hyperparameters and experimental results look like if such a methodology were employed? Demonstrating this would significantly enhance confidence in the method's practical utility.

3. In real-world auditing practices, do auditors commonly rely on Feature Visualization to assess models?

**Ethical Concerns:**

["NO or VERY MINOR ethics concerns only"]

**Final Justification:**

I thank the authors for their response. The rebuttal is promising, but I believe that the paper will require substantial changes to fully address concerns. My score will remain the same.

**Limitations:**

Yes, the paper acknowledges limitations. Depending on the response to Question 3, the authors might consider engaging with real-world auditors to make them aware of the paper findings..

**Quality:**

2

**Strengths And Weaknesses:**

Strengths:
- The visual aspects of the paper are strong, with well-presented figures.
- The underlying concept of the attack appears theoretically sound.
- The authors made an effort to provide theoretical justification for their approach.
- The supplementary information in the appendix was a useful addition.

Weaknesses:

The reviewer has significant concerns about the paper's experimental rigor and the validity of its claims, stemming from several key issues:

1. Lack of Comparative Analysis: The paper fails to compare GS against any existing baselines. While it suggests that methods like Nanfack et al. might degrade model performance, no empirical evidence is provided to substantiate this claim. This omission makes it difficult to assess GS's true effectiveness.

2. Unconvincing Similarity Metrics: The reported similarity metrics (e.g., MSE and SSIM) between the manipulated FV and the target image do not show statistically significant improvement. For instance, these metrics remain identical for alpha values between 1.0 and 0.4, despite a notable degradation in model performance (from 76% to 67%) within that range. This raises questions about the practical impact of the proposed attack.

3. Apparent Cherry-Picking of Results: The selection of specific alpha values in Table 1 (e.g., 0.64 and 0.62, which are very close to 0.6) while omitting results for 0.7 or 0.8 suggests a selective presentation of data. This "cherry-picking" makes it challenging to understand the full landscape of hyperparameter effects and raises doubts about the robustness of the findings.

4. Excessive Hyperparameter Tuning: The wide range and extreme precision of `alpha` values presented in the appendix (e.g., 0.8, 0.025, 0.01, 0.995, 0.64, 0.9995, 0.99991) strongly indicate a reliance on extensive grid searching to find optimal parameters. If the method is truly generalizable, such extreme fine-tuning should not be necessary. This undermines confidence in the method's practical applicability.

5. Unproven Efficacy on Transformers: Despite being a purported advantage over prior work, the paper does not convincingly demonstrate the methodology's effectiveness on transformer architectures. The extremely high alpha values used for ViT models (e.g., at least 0.9995) suggest that the method may simply be injecting noise, rather than performing a targeted attack.

Threat Model Plausibility:
The paper is framed as a security attack, but the reviewer questions the realism of the proposed threat model. Unmanipulated FVs are typically abstract. It is unlikely that an auditor would solely rely on these abstract visualizations without incorporating other metrics or auditing procedures. This diminishes the practical significance of the described attack.

Presentation and Quality:
The paper exhibits some signs of being rushed or lacking polish. For example, Table 2 suffers from poor formatting, indicative of a general lack of attention to detail.

---

> ### Author Rebuttal · Authors · 2025-07-31
>
> We thank the reviewer for their thoughtful and constructive feedback. We are glad that they found the core idea of our method theoretically sound, appreciated the visual presentation and figures, and found the supplementary material helpful. Below, we address the reviewer’s concerns in detail, providing additional experimental results, methodological details, and elaborating on the practical applicability of our method.
>
> ### Weaknesses
>
> > 1. Lack of Comparative Analysis: The paper fails to compare GS against any existing baselines. While it suggests that methods like Nanfack et al. might degrade model performance, no empirical evidence is provided to substantiate this claim. […]
>
> We appreciate the reviewer’s concern about the lack of empirical comparison with prior work. While conceptual differences were discussed in Appendix A, we now include a direct qualitative and quantitative comparison with the only comparable fine-tuning-based method: Prox-Pulse (PP) [Nanfack et al., 2024]. This comparison uses a ResNet-50 setting from Sec. 4 (target image: Dalmatian; manipulated neuron: wolf spider). We follow PP’s reported hyperparameters and additionally vary its $\alpha$ to show the trade-off between model accuracy and manipulation result.
>
> Alongside original similarity metrics, we introduce two new semantic alignment measures:
>
> - Target Label Alignment ("Target Lbl."): CLIP cosine similarity between a FV and the target label embeddings (e.g., “an image of a Dalmatian”).
> - GT Label Alignment ("GT Lbl."): CLIP similarity between a FV and the ground-truth feature label (e.g., “an image of a wolf spider”).
>
> Results (Table A) show that Gradient Slingshots (GS) yield FVs more semantically aligned with the target image and less aligned with the ground truth label, while preserving model performance better than Prox-Pulse. GS outperforms Prox-Pulse on all similarity metrics except low-level metrics MSE and SSIM.
>
> **Table A: Comparison of FV manipulation results on the “wolf spider” output neuron in ResNet-50 between GS and Prox-Pulse (PP).**
>
> | \$\alpha\$ | AUROC |    Acc↑   |  Target Lbl.↑ |    GT Lbl.↓   |     CLIP↑     |      MSE↓     |     LPIPS↓    |   SSIM↑   |
> | :--------: | :---: | :-------: | :-----------: | :-----------: | :-----------: | :-----------: | :-----------: | :-------: |
> |  Original  |  1.00 | **76.13** |   0.23±0.01   |   0.28±0.02   |   0.53±0.02   |   0.12±0.01   |   0.69±0.01   | 0.05±0.01 |
> |   PP 1e−5  |  1.00 |   69.03   |   0.24±0.01   |   0.24±0.01   |   0.53±0.01   | **0.08±0.00** |   1.07±0.03   | **0.08±0.00** |
> |   PP 0.1   |  1.00 |   58.51   |   0.22±0.01   |   0.31±0.01   |   0.54±0.01   |   0.12±0.01   |   0.70±0.01   | 0.05±0.01 |
> |   GS 0.64  |  1.00 |   75.13   | **0.31±0.01** | **0.23±0.01** | **0.68±0.02** |   0.12±0.01   | **0.59±0.02** | 0.04±0.01 |
>
>
> > 2. Unconvincing Similarity Metrics: The reported similarity metrics (e.g., MSE and SSIM) between the manipulated FV and the target image do not show statistically significant improvement. […]
>
> We thank the reviewer for raising the concern. Low-level metrics (MSE, SSIM) are more useful in settings with low-res datasets (e.g., MNIST, CIFAR), where manipulated FVs resemble targets pixel-wise more closely (as noted in Section 4 and Appendix B). In high-res settings, these metrics fail to capture compositional or semantic alignment (see Fig. 3). CLIP similarity better reflects high-level alignment but may still disregard compositional structure.
>
> To further support our evaluation results, we added multiple samples of manipulated FVs per evaluation. We also extended all quantitative evaluations using Target/GT Label Alignment metrics described above. These metrics explicitly quantify conceptual alignments that are invisible to traditional similarity scores. Some of the results using these metrics are shown in Tables A, B, C in this rebuttal.
>
> > 3. Apparent Cherry-Picking of Results: The selection of specific alpha values in Table 1 (e.g., 0.64 and 0.62, which are very close to 0.6) while omitting results for 0.7 or 0.8 suggests a selective presentation of data. […]
>
> The $\alpha$ values in Table 1 were chosen to show representative trade-off states between model performance and FV manipulation success. Within 0.64-0.9, metrics vary marginally, so we did not report these values.
>
> To address the concern, we have now extended Tab. 1 with $\alpha=0.7$ and $0.8$ (see Table B), providing a full sweep from 0.1 to 0.9. These additional results confirm the trends described in Section 4.1. Additionally, in response to reviewer PQ6K, we provide quantitative evidence showing that the neuron targeted for manipulation itself was not cherry-picked by providing evaluation results for 10 more neurons in the ResNet-50 setting. We hope this clarifies our reporting and alleviates concerns about selective presentation.
>
> **Table B: Accuracy–manipulation trade-off for the GS attack on the “wolf spider” output neuron in ResNet-50 (partial Tab. 1).**
>
> | \$\alpha\$ |    AUC   |  Acc. |  Target Lbl.↑  |    GT Lbl.↓    |   CLIP↑   |    MSE↓   |   LPIPS↓  |   SSIM↑   |
> | :--------: | :------: | :---: | :-------: | :-------: | :-------: | :-------: | :-------: | :-------: |
> |     0.8    | 1.00 | 76.00 | 0.22±0.01 | 0.28±0.02 | 0.53±0.02 | 0.12±0.01 | 0.69±0.01 | 0.05±0.01 |
> |     0.7    | 1.00 | 75.77 | 0.23±0.02 | 0.27±0.02 | 0.53±0.03 | 0.13±0.01 | 0.67±0.02 | 0.05±0.01 |
>
>
> > 4. Excessive Hyperparameter Tuning: The wide range and extreme precision of alpha values […] strongly indicate a reliance on extensive grid searching to find optimal parameters. […]
>
> We thank the reviewer for raising this concern. A detailed description of our hyperparameter tuning and selection procedure is provided in our response to Question 2 below, and we have incorporated this explanation into the revised paper.
>
> > 5. Unproven Efficacy on Transformers: Despite being a purported advantage over prior work, the paper does not convincingly demonstrate the methodology's effectiveness on transformer architectures. The extremely high alpha values used for ViT models (e.g., at least 0.9995) suggest that the method may simply be injecting noise, rather than performing a targeted attack.
>
> We thank the reviewer for the observation. $\alpha$ parameter balances manipulation and preservation losses; high $\alpha$ does not indicate noise injection but rather that the manipulation loss remains small relative to activation changes during training.
>
> Figure 1 and Section 5 demonstrate strong efficacy of GS on a ViT model (CLIP ViT-L/14). While quantitative metrics for ViT-L/32 (Appendix D) are less convincing than the qualitative results (Fig. 3), this highlights the limitations of conventional metrics in capturing similarity to the target. To address this, we extended the evaluation with the two new alignment metrics introduced above. Results in Table C show that for certain $\alpha$ values, manipulated FVs align better with the target label (“sea lions”) than the ground truth (“broccoli”), sometimes even surpassing the original FV’s alignment with its ground truth label.
>
> **Table C: Accuracy–manipulation trade-off for the GS attack on the ``broccoli'' output neuron in ViT-L/32 (partial Tab. 16).**
>
> | \$\alpha\$ |    Acc.   |  Target Lbl. ↑  |    GT Lbl. ↓    |
> | :--------: | :-------: | :-------------: | :-------------: |
> |  Original  | **76.97** |   0.22 ± 0.03   |   0.28 ± 0.02   |
> |  0.999999  |   76.58   |   0.23 ± 0.02   |   0.28 ± 0.02   |
> |   0.9999   |   76.64   | **0.29 ± 0.02** |   0.27 ± 0.03   |
> |   0.9995   |   76.06   | **0.29 ± 0.02** | **0.25 ± 0.01** |
> |    0.999   |   75.96   |   0.28 ± 0.02   | **0.25 ± 0.01** |
> |    0.99    |   72.42   | **0.29 ± 0.02** | **0.25 ± 0.01** |
> |     0.1    |    0.1    |   0.25 ± 0.01   |   0.26 ± 0.01   |
>
> ### Questions
>
> > 1. Could the authors clarify the meaning of the standard deviations reported throughout the paper? […]
>
> As explained in Appendix C.8, the standard deviations reflect variability across multiple FV generations—100 per experiment (30 for CLIP ViT-L/14 due to compute limits). Since FV generation involves sampling initializations, each run differs. The deviations thus capture metric consistency across independent runs. We’ve clarified this in all relevant table captions.
>
> > 2. Is there a methodology for determining reasonable hyperparameter values without resorting to an exhaustive grid search? […]
>
> Our tuning of the parameter $\alpha$ uses a coarse-to-fine iterative approach rather than an exhaustive grid search. We start with a broad sweep (e.g., $\alpha \in \[0.01, 0.05, 0.5, 0.9\]$) to identify promising regions, then refine around top candidates (e.g., testing values near 0.5), iteratively narrowing down.
>
> Importantly, tuning is done using only 1–3 epochs on small subsets of large datasets (see Table 8 in the Appendix). For ViT experiments, this involves under 1\% of ImageNet and takes less than 5 minutes per epoch, making the process practical even on modest resources.
>
> For parameter $w$, we set $w=0.1$ when the target layer has 10 neurons, and $w=0.01$ when it has 1000 neurons, to avoid excessive disruption of the activation landscape of the feature. For $\gamma$, we use the heuristic $\nabla (\phi \circ \eta)(\mathbf{0}) = \gamma \mathbf{q^t}$ to preserve the gradient magnitude in the initialization zone.
>
> We have incorporated this explanation into the paper.
>
> > 3. In real-world auditing practices, do auditors commonly rely on FV to assess models?
>
> While FV is not yet standard in traditional audits, it has proven useful for detecting undesirable behaviors (Goh et al., 2021), uncovering backdoor attacks (Casper et al., 2023), interpreting time-series patterns (Schlegel et al., 2024), and understanding CNN filters in materials science (Ling et al., 2017; Zhong et al., 2022). These show FV’s value in the broader context of model interpretability and safety.

---

> > ### Comment · Reviewer_Tk8Z · 2025-08-05
> >
> > I thank the authors for their response. The rebuttal is promising, but I believe that the paper will require substantial changes to fully address concerns. My score will remain the same.

---

> > > ### Author Response · Authors · 2025-08-05
> > >
> > > We thank the reviewer for their thoughtful engagement and are pleased to hear that the rebuttal was found promising. While our response is extensive, the additions are purely supportive in nature as they provide further empirical validation and address specific concerns (e.g., cherry-picking), but do not require or result in substantial changes to the paper. The core method, main claims, and overall scope of the paper remain exactly as originally submitted.
> > >
> > > Concretely, we:
> > > - Added experimental results to support the existing methodological comparison with Prox-Pulse to illustrate differences in performance preservation and manipulation strength;
> > > - Added new metrics (Target/GT Label Alignment) to complement the similarity metrics already reported in the paper, better capturing high-level consistency in high-resolution settings, including those involving ViTs;
> > > - Reported results for additional hyperparameter values and neurons, and included further examples of manipulated FVs, to enhance completeness and demonstrate that results were not cherry-picked;
> > > - Expanded the commentary on hyperparameter selection and further clarified the meaning of reported standard deviations.
> > >
> > > We thank the reviewer once again for raising these points, which helped us improve the clarity and completeness of the paper.

---

> > > ### Author Response · Authors · 2025-08-06
> > >
> > > After further considering your latest comment, we realized that in our previous response, we missed the opportunity to ask which of your concerns remained unaddressed. We are committed to improving the paper, and while we're encouraged that you found our rebuttal “promising”, you also mentioned the need for “substantial changes”. Could you kindly clarify what additional changes you recommend beyond those we've already addressed?

---

### Note · Authors · 2025-08-13

We sincerely thank the reviewers for their thoughtful feedback, which enabled us to refine our work. Building on our original theoretical, methodological, and experimental framework, we have added decisive new evidence confirming that our *Gradient Slingshots* attack is effective, robust, and broadly applicable.

We would also like to thank the reviewers for their positive assessment of our contribution, including its *novelty* (PQ6K, zpF6, brRT), *theoretical soundness* (Tk8Z), *comprehensive empirical validation* (brRT, zpF6), *quality of writing* (PQ6K, zpF6), and *strong visual aspects* (Tk8Z), with PQ6K calling our work "interesting" and "elegant". The majority of reviewers expressed a positive recommendation.

**Key revisions in response to reviewer concerns:**
- **Baselines** (App. A): Demonstrated that Gradient Slingshots outperforms Prox-Pulse in manipulation success and performance preservation.
- **Metrics and transformer results** (Sec. 4.1, App. D.1): Clarified limitations of conventional similarity metrics in our high-resolution settings; introduced *Target/GT Label Alignment* metrics, confirming that manipulated Feature Visualizations (FVs) align closely with target rather than ground-truth labels, including for ViTs.
- **Robustness confirmed** (App. D): Expanded evaluation across hyperparameters and neurons, included additional FV examples, demonstrating that results are stable, not cherry-picked, and showing the attack does not affect non-manipulated features.
- **Hyperparameter selection** (App. C): Expanded commentary on our hyperparameter selection procedure.
- **Broader applicability** (App. D): Demonstrated attack efficacy on MACO, showing applicability across FV variants.
- **Presentation and clarity**: Moved evaluation metrics description to the main text, relocated Lemma 3.1 proof to the appendix, corrected typos in Equations 4 and 10, and clarified the meaning of reported standard deviations.

Reviewers recognized these improvements: zpF6 confirmed "all concerns" were resolved, PQ6K acknowledged progress on their points, and Tk8Z called the rebuttal "promising."

Our work reveals a fundamental vulnerability in Feature Visualization, a core interpretability tool, raising urgent concerns about its reliability. Given the growing regulatory and community emphasis on AI safety and transparency, and the central role of interpretability in trustworthy AI, we believe this contribution is both timely and highly impactful for NeurIPS.

---

### Decision · Program_Chairs · 2025-09-17

**Decision:**

Accept (poster)

**Comment:**

This paper introduces Gradient Slingshots (GS) to attack the Feature Visualization method. GS manipulates a model to conceal malicious intent, without significantly hurting the performance of the original model. The novelty and effectiveness of the proposed method are praised by reviewers. However, a more thorough explanation regarding whether the method injects noises is suggested. Overall, this paper meets the publication standards.